

# Using GECKO-A to derive mechanistic understanding of SOA formation from the ubiquitous but understudied camphene

Isaac Kwadjo Afreh[1], Bernard Aumont[2], Marie Camredon[2], Kelley Claire Barsanti[1]

[1]Department of Chemical and Environmental Engineering and College of Engineering-Center for Environmental Research and Technology (CE-CERT), University of California-Riverside, Riverside, California 92507, United States
[2]LISA, UMR CNRS 7583, Université Paris-Est-Créteil, Université de Paris, Institut Pierre Simon Laplace, Créteil, France

*Correspondence to*: Kelley Barsanti (kbarsanti@engr.ucr.edu)

**Abstract**

Camphene, a dominant monoterpene emitted from both biogenic and pyrogenic sources, has been significantly understudied, particularly in regard to secondary organic aerosol (SOA) formation. When camphene represents a significant fraction of emissions, the lack of model parameterizations for camphene can result in inadequate representation of gas-phase chemistry and underprediction of SOA formation. In this work, the first mechanistic study of SOA formation from camphene was performed using the Generator for Explicit Chemistry and Kinetics of Organics in the Atmosphere (GECKO-A). GECKO-A was used to generate gas-phase chemical mechanisms for camphene and two well-studied monoterpenes, α-pinene and limonene; and to predict SOA mass formation and composition based on gas/particle partitioning theory. The model simulations represented observed trends in published gas-phase reaction pathways and SOA yields well under chamber-relevant photooxidation and dark ozonolysis conditions. For photooxidation conditions, 70 % of the simulated α-pinene oxidation products remained in the gas phase compared to 50 % for limonene; supporting model predictions and observations of limonene having higher SOA yields than α-pinene under equivalent conditions. The top 10 simulated particle-phase products in the α-pinene and limonene simulations represented 37-50 % of the SOA mass formed and 6-27 % of the hydrocarbon mass reacted. To facilitate comparison of camphene with α-pinene and limonene, model simulations were run under idealized atmospheric conditions, wherein the gas-phase oxidant levels were controlled. Metrics for comparison included: gas-phase reactivity profiles, time-evolution of SOA mass and yields, and physicochemical property distributions of gas- and particle-phase products. The controlled-reactivity simulations demonstrated that: (1) in the early stages of oxidation, camphene is predicted to form very low volatility products, lower than α-pinene and limonene, which condense at low mass loadings; and (2) the final simulated SOA yield for camphene (46 %) was relatively high, in between α-pinene (25 %) and limonene (74 %). A 50/50 (α-pinene/limonene) mixture was then used as a surrogate to represent SOA formation from camphene; while simulated SOA mass and yield were well represented, the volatility distribution of the particle-phase products was not. To demonstrate the potential importance of including a parameterized representation of SOA formation by camphene in air quality models, SOA mass and yield were predicted for three wildland fire fuels based on measured monoterpene distributions, and published SOA parameterizations for α-pinene and limonene. Using the 50/50 surrogate mixture to represent camphene increased predicted SOA mass by 43-50 % for black spruce and by 56-108 % for Douglas fir. This first detailed modeling study of the gas-phase oxidation of camphene and subsequent SOA formation provides an opportunity for future measurement-model comparisons and lays the foundation for developing chemical mechanism and SOA parameterizations for camphene that are suitable for air quality modeling.



# 1 Introduction

Sources of atmospheric monoterpene ($C_{10}H_{16}$) emissions are diverse, and include biogenic sources (Geron et al., 2000; Guenther et al., 1995; Hayward et al., 2001; Kesselmeier and Staudt, 1999; Kim et al., 2010; Ludley et al., 2009; Maleknia et al., 2007; Rinne et al., 2000; Steinbrecher et al., 1999; Tani et al., 2003; White et al., 2008), as well as pyrogenic sources (Akagi et al., 2011, 2013; Gilman et al., 2015; Hatch et al., 2015; Simpson et al., 2011). Monoterpenes account for an estimated one-fifth of total global biogenic volatile organic compound (BVOC) emissions (Guenther et al., 1995; Hallquist et al., 1999). Quantities and identities of monoterpenes emitted from biogenic sources primarily depend on plant species and temperature/light (Geron et al., 2000; Hayward et al., 2001; Yáñez-Serrano et al., 2018). Studies across biogenic source types (e.g., terrestrial vegetation, soil, and marine) typically include up to 14 individual monoterpenes, with α-pinene, β-pinene, camphene, 3-carene, limonene, myrcene, p-ocimene, and sabinene being the most widely reported and having the highest emissions (Ambrose et al., 2010; Bäck et al., 2012; Fehsenfeld et al., 1992; Geron et al., 2000; Hayward et al., 2001; Rinne et al., 2000; White et al., 2008; Yassaa et al., 2008). As with biogenic sources, the identities and quantities of monoterpenes from pyrogenic sources (e.g., biomass burning) vary as a function of plant species and fuel component (Hatch et al., 2019). Approximately 30 monoterpene isomers have been observed from biomass burning sources, with α-pinene, β-pinene, camphene, 3-carene, limonene, and myrcene being commonly detected (Akagi et al., 2013; Gilman et al., 2015; Hatch et al., 2015).

Monoterpenes have a wide range of molecular structures, atmospheric lifetimes, and secondary organic aerosol (SOA) formation potentials. The molecular structures of monoterpenes can be acyclic or cyclic (with variability in the size and number of rings) and can include one to three C=C double bonds (Atkinson and Arey, 2003b; Calogirou et al., 1999; Jacobson et al., 2000; Lee et al., 2006a). The reaction rates of monoterpenes with atmospheric oxidants vary by orders of magnitude (Atkinson and Arey, 2003a; Geron et al., 2000); consequently, their atmospheric lifetimes vary from minutes to days (Atkinson and Arey, 2003b). Monoterpenes can react with atmospheric oxidants to form less-volatile oxidation products leading to the formation of SOA. SOA composes a significant fraction of atmospheric fine particulate matter ($PM_{2.5}$), which adversely affects air quality and impacts climate (Almatarneh et al., 2018; Hallquist et al., 1999; Jacobson et al., 2000; Kanakidou et al., 2004). The extent of SOA formation from monoterpenes can vary significantly, due to differences in their structures, reaction rates, and volatility of their oxidation and accretion products (Griffin et al., 1999; Ng et al., 2007; Zhang et al., 2015).

Over the past two decades, chamber studies have been performed using monoterpene precursors to elucidate their potential to form SOA under conditions approximating atmospheric relevance. For example, Griffin et al. (1999) used a series of outdoor chamber experiments to establish the SOA formation potential of fourteen biogenic compounds, including nine monoterpenes. Since then, several chamber studies under varying experimental conditions have been conducted for individual monoterpenes including α-pinene, β-pinene, 3-carene, limonene, and myrcene (e.g., Amin et al., 2013; Boyd et al., 2017; Fry et al., 2014; Hatfield and Huff Hartz, 2011; Lee et al., 2006a; Ng et al., 2007; Presto et al., 2005; Presto and Donahue, 2006; Zhao et al., 2018). Additionally, chamber studies have been conducted to investigate gas-phase reaction pathways and major products from the reactions of monoterpenes with hydroxyl radical (OH), ozone ($O_3$), and nitrate radical ($NO_3$) (e.g., Draper et al., 2015; Kundu et al., 2012; Zhang et al., 2015). While some monoterpenes have been well studied in chambers or other laboratory reactors, other monoterpenes are relatively under studied, including some that are commonly measured in non-negligible quantities in the atmosphere.

Parameterizations used in air quality models are largely based on laboratory studies, thus widely studied monoterpenes (e.g., α-pinene and limonene) are often used as surrogates to represent the gas-phase chemistry and SOA formation of all terpenes (e.g., Carter, 2010; Saha and Grieshop, 2016; Stockwell et al., 1997). The lack of monoterpene-specific laboratory data can result in inadequate representation of monoterpene chemistry, including SOA formation, particularly where a diversity of unrepresented





monoterpenes has a large contribution to total emissions. Camphene is one monoterpene that has been observed in the atmosphere but has little to no published data regarding SOA formation. Previous experimental and theoretical studies of camphene focused on the gas-phase reactions of camphene and product identification (Atkinson et al., 1990; Gaona-Colmán et al., 2017; Hakola et al., 1994). Recently, a density functional theory (DFT) approach was also used to investigate the oxidation of camphene and the fate of product radicals under atmospherically relevant conditions (Baruah et al., 2018). While this approach identified plausible

reaction pathways of camphene photooxidation and associated gas-phase products, formation of SOA was not considered.

In this work, a mechanistic study of SOA formation from camphene was conducted using the Generator for Explicit Chemistry and Kinetics of Organics in the Atmosphere (GECKO-A). GECKO-A has been previously used to study SOA formation from a number of precursors (e.g., Camredon et al., 2007; La et al., 2016; McVay et al., 2016; Valorso et al., 2011). GECKO-A was used here to generate nearly explicit mechanisms for camphene and the well-studied monoterpenes α-pinene and limonene.

Model simulations were run under chamber-relevant conditions ("chamber reactivity simulations") to capture trends in simulated SOA mass and composition and compare with published observations using commonly reported metrics including SOA yields and oxygen/carbon (O/C) ratios. Model simulations were also run under idealized atmospheric conditions ("controlled reactivity simulations") to perform a direct comparison of  camphene with α-pinene and limonene under tropospheric conditions; including comparisons of gas-phase oxidation pathways, gas-phase reactivity profiles, time-evolution of SOA mass and yields, and

physicochemical property distributions of gas- and particle-phase products. The feasibility of using α-pinene or limonene as a surrogate for camphene was assessed. Based on these analyses, implications for air quality model predictions and opportunities for future studies were identified.

## 2 Method

### 2.1 GECKO-A Model description

SOA formation from three monoterpene precursors (α-pinene, limonene, and camphene) was modeled using GECKO-A. A description of GECKO-A is given by Aumont et al. (2005). GECKO-A is a modeling tool that generates nearly explicit gas-phase oxidation mechanisms for individual or multiple organic compounds under general atmospheric conditions (Aumont et al., 2005, 2012; Camredon et al., 2007), as well as the properties to represent the gas/particle mass transfer of the stable organic compounds present in the mechanisms (Camredon et al., 2007; Valorso et al., 2011). The nearly explicit chemical mechanism is

generated using experimental data and a predefined protocol (Aumont et al., 2005, 2012; Camredon et al., 2007). The protocol is described in Aumont et al. (2005) and includes updates described in Aumont et al. (2013), La et al. (2016), McVay et al. (2016), and Valorso et al. (2011). In the absence of experimental data, reaction rate constants and products, as well as their physicochemical properties, are estimated based on structure-activity relationships (SARs) (Aumont et al., 2005). The saturation vapor pressures of stable organic compounds were estimated in this work using the Nannoolal method (Nannoolal et al., 2008). Autoxidation,

however, is not considered in the current version of GECKO-A.

Some simplifications were applied in this work during the mechanism generation to reduce the size of the gas-phase chemical mechanisms: (1) the maximum generations of oxidation for each mechanism was set at six based on prior GECKO-A modeling results, where increasing the number of generations beyond six did not result in significant changes in the evolution of the gas and particle phases (Aumont et al., 2012); (2) species with vapor pressure below $10^{-13}$ atm were considered to partition

completely to the particle phase and therefore treated as end products during gas-phase mechanism generation (Valorso et al., 2011); (3) position isomers were lumped if the production yield of a species was lower than $10^{-3}$ (Valorso et al., 2011). The chemical mechanisms generated for this study included: $1.4 \times 10^6$ reactions and $2 \times 10^5$ oxidation products for α-pinene; $6.5 \times 10^5$



reactions and $9.3 \times 10^4$ oxidation products for limonene; and $1.3 \times 10^6$ reactions and $1.8 \times 10^5$ oxidation products for camphene. These mechanisms were then implemented in a box model to simulate the evolution of gaseous organic compounds and SOA

formation (Aumont et al., 2005, 2012; Camredon et al., 2007). In the version of GECKO-A used for this work, gas/particle partitioning was calculated according to the saturation vapor pressure of each organic compound and assuming thermodynamic equilibrium between the gas and an ideal (activity coefficients = 1), homogeneous, and inert condensed phase. No condensed-phase reactions were included.

## 2.2 GECKO-A generated oxidation mechanisms

### 2.2.1 OH reaction scheme

The reaction pathways of OH-initiated oxidation of α-pinene, limonene, and camphene up to the formation of 1st-generation stable products are shown in Figs. 1, 2, and 3, respectively. The initial reaction steps proceed mainly by the addition of OH to the C=C double bond or by hydrogen abstraction. This leads to the formation of hydroxyalkyl radicals which react rapidly with $O_2$ to form peroxy radicals. The peroxy radicals can combine with NO, $RO_2$ or $HO_2$ to form stable products. The peroxy

radicals can also lose an oxygen atom through reaction with NO to form alkoxy radicals, which is consistent with observations reported by Atkinson and Arey (1998) and Calogirou et al. (1999). For α-pinene oxidation, the hydroxyalkyl radicals primarily react with $O_2$ to form four-membered ring peroxy radicals, which then react with NO, $RO_2$ or $HO_2$ to form stable products or lose an oxygen atom to form alkoxy radicals. As observed by Lee et al. (2006b), the alkoxy radicals undergo subsequent reactions leading to formation of formaldehyde, acetone, and multifunctional products including pinonaldehyde. For limonene oxidation,

reaction of the peroxy radicals with $NO/NO_3/RO_2$ followed by $O_2$ addition and NO to $NO_2$ conversion leads to the formation of limononaldehyde or limonaketone and formaldehyde, which are consistent with observations reported by Lee et al. (2006b). Alternatively, the peroxy radicals react with $NO/NO_3/RO_2$ to form ring-opened peroxy radicals, which further react to form multifunctional products. For camphene, the hydroxyalkyl radicals react rapidly with $O_2$ to form hydroxyalkylperoxy radicals. The hydroxyalkylperoxy radicals subsequently react with NO, $RO_2$, and $HO_2$ to form stable products, or react with $NO/NO_3/RO_2$ to

form hydroxyalkoxy radicals. The hydroxyalkoxy radicals then either decompose to form camphenoline (a bicyclic product) and formaldehyde, or react with $O_2$ to form five-membered ring hydroxyperoxy radicals, which further react to form multifunctional products. The reaction pathway of OH addition to the exocyclic double bond of camphene as represented in GECKO-A is in agreement with the observations made by Gaona-Colmán et al. (2017) and Reissell et al. (1999), as well as by Baruah et al. (2018) in their DFT study of OH-initiated oxidation of camphene. While camphene and α-pinene are structurally bicyclic, their 1st

generation products resulting from the decomposition of the bicyclic hydroxyalkoxy radicals differ; camphene primarily forms five-membered ring 1st generation products while α-pinene primarily forms four-membered ring 1st generation products. Limonene, which is monocyclic, primarily forms ring-opened 1st generation products when its monocyclic hydroxyalkoxy radicals decompose.

### 2.2.2 $O_3$ reaction scheme

The initial oxidation pathways of $O_3$-initiated oxidation of α-pinene, limonene, and camphene are shown in Figs. S1, S2,

and S3, respectively. The reaction starts with the addition of $O_3$ to the C=C double bond of the parent compound to form an ozonide, which rapidly undergoes bond cleavage to form a biradical Criegee intermediate bearing a carbonyl substituent for terpenes with an endocyclic double bond, or a biradical Criegee intermediate and a carbonyl for terpenes with an exocyclic double bond. The Criegee intermediate can stabilize by collisions and/or decompose (after possible rearrangement) to form peroxy radicals. The stabilized Criegee intermediates (SCI) undergo bimolecular reactions with $H_2O$, CO, NO and/or $NO_2$. The peroxy radicals then





react with $HO_2/NO/RO_2$ to form stable products or react with $NO/NO_3/RO_2$ to form alkoxy radicals. For α-pinene, the alkoxy radicals either react with $O_2$ or decompose to form formaldehyde, an acetaldehyde, and peroxy radicals. The peroxy radicals further react to form peroxy acid, carboxylic acid, and $CO_2$. For limonene, the alkoxy radical reactions primarily lead to the formation of organic nitrates, organic hydroperoxides, carboxylic acids, and peroxy acids. For camphene, the ozonide decomposes to form (1) champhenilone, a stable bicyclic product that has been observed experimentally by Calogirou et al. (1999) and Hakola et al. (1994);

and (2) a bicyclic peroxy radical and formaldehyde, consistent with the camphene + $O_3$ mechanism reported by Gaona-Colmán et al. (2017). The bicyclic peroxy radical reacts with $HO_2/NO/RO_2$ to form stable products or reacts with $NO/NO_3/RO_2$ to form alkoxy radical which then further reacts to form five-membered ring products.

### 2.2.3 $NO_3$ reaction scheme

The initial oxidation pathways of $NO_3$-initiated oxidation of α-pinene, limonene, and camphene are shown in Figs. S4,

S5, and S6, respectively. The $NO_3$ radical attacks the C=C double bond to form a nitratoalkyl radical which undergoes rapid reaction with $O_2$ to form a nitratoalkylperoxy radical. The nitratoalkylperoxy radicals of all three compounds react similarly in three ways: (1) with NO to form dinitrates; (2) with $HO_2$ or $RO_2$ to form nitratocarbonyls, nitratoalcohols, and nitratoperoxides (Calogirou et al., 1999); and (3) with $NO/NO_3/RO_2$ to form nitratoalkoxy radicals, which react further to form multifunctional products.

### 2.3 Simulation conditions

The objective of the chamber reactivity simulations was to compare GECKO-A model output with published SOA chamber data (Table 1). Here, no attempt is made to strictly reproduce the conditions of a given chamber experiment. Since the first objective of this study focuses on the ability of the model to capture the major trends observed in chamber data (e.g., SOA yields and major species), the simulation conditions were therefore set to mimic (or be representative of) typical chamber

conditions. Comparative analyses were performed for the precursors α-pinene and limonene, since they are among the well-studied monoterpenes in environmental chambers and sufficient data exist for measurement-model comparison. These simulations, chamber reactivity simulations, included photooxidation (P) and dark ozonolysis (DO) conditions, which were differentiated by the initial concentrations of NO, HONO, and $O_3$ as shown in Table 1. For both the P and DO conditions, the initial hydrocarbon mixing ratios were set at a relatively low (50 ppb) and a relatively high (150 ppb) level as compared with published chamber

studies. This resulted in a total of four chamber reactivity simulations for each monoterpene precursor. In each simulation, 1 μg m$^{-3}$ of organic seed with molecular weight of 250 g mol$^{-1}$ was added to initiate gas/particle partitioning.

The objective of the controlled reactivity simulations was to examine SOA formation by camphene in the context of well-studied monoterpenes, specifically α-pinene and limonene, under controlled tropospheric conditions (Table 2). In these simulations, the gas-phase chemistry was not controlled by the individual precursors, but by other organic compounds as occurs in the ambient atmosphere. A mixture of ethane (10 ppb) and formaldehyde (50 ppb) was used to buffer (i.e. control) the gas-phase

reactivity. Hence, the levels of the oxidants do not change when relatively small amounts of precursor were added in the simulation and therefore allows a straightforward comparison of oxidation mechanisms of the various terpenes. The $NO_x$ and $O_3$ mixing ratios were held constant at values of 1 ppb and 30 ppb respectively throughout the simulation. The controlled reactivity simulations included 0.1 ppb of initial precursor and 10 μg m$^{-3}$ of organic seed.

All box-model simulations were performed under the following environmental conditions: temperature was fixed at 298 K; humidity was held at 5 %; and the solar zenith angle (required to compute the photolysis frequencies) was set at 50°, except for dark ozonolysis conditions where no photolysis was considered.



## 3 Results

### 3.1 Chamber reactivity simulations

#### 3.1.1 Model-measurement comparison


In Fig. 4, SOA yields from the chamber reactivity simulations are shown with measured SOA yields from chamber studies. SOA data (see Table S1) were compiled from 12 published chamber studies (e.g., Chen et al., 2017; Griffin et al., 1999; Kim and Paulson, 2013; Kourtchev et al., 2014; Ng et al., 2007; Yu et al., 1999) in which α-pinene or limonene was used as a precursor and final SOA mass, SOA yield, and reacted hydrocarbon concentration (ΔHC) were reported (at least two of the three quantities). For

the α-pinene photooxidation data, there is an apparent cluster around an SOA yield of 0.2 for SOA mass < 150 µg m$^{-3}$ with which the model agrees (Fig. 4a). The scatter in the data is due to varying experimental conditions (e.g., temperature and NO$_x$ mixing ratio). As previously observed, SOA yields of α-pinene tend to be higher at lower temperatures and lower NO$_x$ conditions (higher initial VOC/NO$_x$ ratios) (Kim and Paulson, 2013; Pathak et al., 2007b). For example, the two relatively high SOA yields (0.38 at 29.3 µg m$^{-3}$ and 0.46 at 121.3 µg m$^{-3}$) had relatively low initial NO$_x$ concentrations (Ng et al., 2007), while the two relatively low

SOA yields (0.059 at 44 µg m$^{-3}$ and 0.06 at 4.5 µg m$^{-3}$) had relatively high initial NO$_x$ concentrations (Kim and Paulson, 2013; Ng et al., 2007). For mass loadings > 150 µg m$^{-3}$, α-pinene photooxidation SOA yield data plateaus at approximately 0.3, which also is captured by the model. In contrast, for limonene photooxidation, experimental data show a linear trend in the SOA yield as a function of SOA mass (for SOA mass > 25 µg m$^{-3}$), and the SOA yield does not plateau at higher SOA mass loadings. The observed linear trend in SOA yield as a function of SOA mass is reflected in the model simulations (Fig. 4c). For α-pinene ozonolysis (Fig.

4b), there is an apparent cluster around an SOA yield of 0.2 for SOA mass < 200 µg m$^{-3}$ with which the model agrees. The SOA yield plateaus at approximately 0.4 for SOA mass > 200 µg m$^{-3}$; the model simulations do not extend to this high mass range. For limonene ozonolysis, Fig. 4d shows the chamber SOA yield plateauing at approximately 0.8 (for mass loadings > 200 µg m$^{-3}$) which is captured by the model simulations. Overall, the model agrees well with the observed trends in SOA yield as a function of SOA mass.


Table 3 shows the simulated SOA mass-weighted average oxygen/carbon (O/C) ratios for α-pinene photooxidation (O/C = 0.93), α-pinene ozonolysis (O/C = 0.64), limonene photooxidation (O/C = 0.96), and limonene ozonolysis (O/C = 0.68). For α-pinene, the simulated SOA from photooxidation had higher average O/C than from ozonolysis. This is consistent with experiments by Kourtchev et al. (2015) in which the reported O/C for OH-initiated α-pinene SOA was higher than for α-pinene SOA initiated by ozonolysis. The same trend was predicted for limonene. Generally, the simulated O/C values were high relative to values

reported from chamber studies. Reported average O/C values from chamber studies range from 0.3 to 0.65 for α-pinene photooxidation (e.g., Lambe et al., 2015; Pfaffenberger et al., 2013), 0.22 to 0.55 for α-pinene ozonolysis (e.g., Chen et al., 2011; Chhabra et al., 2010; Kourtchev et al., 2015), and 0.23 to 0.5 for limonene ozonolysis (e.g., Draper et al., 2015; Heaton et al., 2007; Walser et al., 2008). Factors known to affect the O/C ratios include mass loading, OH exposure (defined as the integral of OH concentration and residence time (Lambe et al., 2015)), and oligomerization. Shilling et al. (2009) showed the dependency of O/C

ratios on mass loadings, in which O/C ratio decreased from 0.45 to 0.38 as mass loading increased from 0.5 to 15 µg m$^{-3}$. Mass loading is not likely driving the differences in simulations and observations here, since the simulated mass loadings were similar to the mass loadings of the chamber experiments (e.g., Chhabra et al., 2011; Shilling et al., 2009) with which the O/C ratios were compared. Regarding OH exposure, calculated OH exposures for the photooxidation simulations (Table 3) were within the typically reported OH exposure ranges (5.4×10$^{10}$–4.0×10$^{11}$ molec cm$^{-3}$ s) from the chamber photooxidation experiments (e.g.,

Lambe et al., 2015; Pfaffenberger et al., 2013). Therefore, one explanation for the lower observed O/C values is the loss of H$_2$O during oligomerization (Chhabra et al., 2010; Reinhardt et al., 2007), a process that was likely occurring in the experiments but





was not represented in the GECKO-A simulations. Additionally, for dark ozonolysis, the OH scavengers typically present in chamber experiments (e.g., Kourtchev et al., 2015; Shilling et al., 2009) were absent from the simulations.

### 3.1.2 Major products simulated for α-pinene and limonene

The results from the simulations using the lower hydrocarbon mixing ratio (LHC) and higher hydrocarbon mixing ratio (HHC) were qualitatively similar. Thus, here and in subsequent sections, only the results for the LHC simulations are shown and discussed; the corresponding figures for the HHC simulations are provided in the supplement. Figures 5a and 5b show the chemical structures and molecular formulae of the top 10 products in the gas and particle phases from the α-pinene photooxidation simulation. The top 10 gas-phase products (dominated by carbonyl, carboxyl, and nitrate functional groups) account for 46 % of

the reacted α-pinene carbon mass, with acetone being the top contributor. Two of the top 10 gas-phase products, pinonic acid (i.e. (3-acetyl-2,2-dimethylcyclobutyl)acetic acid) and pinonaldehyde (i.e. (3-acetyl-2,2-dimethylcyclobutyl)acetaldehyde) are among the most commonly reported products in experimental studies (e.g., Lee et al., 2006b). The top 10 particle-phase products (dominated by carbonyl, carboxyl, hydroxyl, hydroperoxide, and nitrate functional groups) account for 42 % of the SOA mass and 7 % of the reacted α-pinene carbon mass. For limonene photooxidation (Figs. S10 and S11), the top 10 gas-phase products account

for 34 % of reacted limonene, while the top 10 particle-phase products account for 50 % of the SOA mass and 20 % of the reacted limonene carbon mass. The top 10 particle-phase products are dominated by dinitrate and carbonyl functional groups, indicating the possible influence of multigeneration products from peroxy radicals + NO reactions.

The top 10 gas- and particle-phase products from the α-pinene ozonolysis simulation are shown in Figs. 6a and 6b. The top 10 gas-phase products account for 62 % of the reacted α-pinene carbon mass, while the top 10 particle-phase products account

for 42 % of the SOA mass and 6 % of the reacted α-pinene carbon mass. Three of the top 10 products have been previously reported in experimental product studies of α-pinene ozonolysis (e.g., Jang and Kamens, 1999; Larsen et al., 2001; Yu et al., 1999). They include one particle-phase product, pinic acid (3-(carboxymethyl)-2,2-dimethylcyclobutane-1-carboxylic acid); and two gas-phase products, pinonic acid (i.e. (3-acetyl-2,2-dimethylcyclobutyl)acetic acid) and pinonaldehyde (i.e. (3-acetyl-2,2-dimethylcyclobutyl)acetaldehyde). For limonene ozonolysis (Figs. S12 and S13), the top 10 gas-phase products account for 24 %

of reacted limonene, while the top 10 particle-phase products account for 37 % of the SOA mass and 27 % of the reacted limonene carbon mass. The top 10 particle-phase products were dominated by carbonyl, carboxyl, hydroxyl, and hydroperoxide, indicating the influence of multi-generational products via peroxy radicals + $HO_2/RO_2$.

### 3.1.3 Modeled SOA yield and carbon budget

Given the skill of the model in representing published chamber data (at both microscopic and molecular levels), the model

was used to explore the carbon budget in the simulations during photooxidation and ozonolysis. The time evolution of SOA yields for α-pinene and limonene during photooxidation and ozonolysis, as simulated by GECKO-A, is shown in Figs. 7a and 7b respectively. Also shown are the corresponding final SOA mass concentrations. As has been previously reported (Lee et al., 2006b), limonene had a higher SOA yield than α-pinene under both photooxidation and ozonolysis conditions.

The time evolution of the carbon budget during the photooxidation and ozonolysis simulations is shown in Figs. 7c to 7f.

During photooxidation (Fig. 7c), the precursors were oxidized largely by OH and $O_3$ (see Fig. S7 for the relative fractions of precursor reacting with each oxidant), forming organic oxidation products in the gas phase. These gaseous oxidation products partitioned into the particle phase if their volatility was low enough. Oxidation products that remained in the gas phase reacted with OH, $NO_3$, and/or $O_3$, or were photolyzed if a chromophore was present; these subsequent gas-phase reactions formed additional oxidation products that partitioned to the particle phase or continued to react in the gas phase. At the end of 12 hours of





photooxidation, the α-pinene system was dominated by organic oxidation products in the gas phase (70 %), with the remaining
        fractions being organic oxidation products in the particle phase (8 %) and $CO+CO_2$ (22 %). The high yield of gas-phase organics
        is largely influenced by the high concentrations of acetone and volatile C8 to C10 species (see Fig. 5a for top gas-phase products
        and Fig. S9a for the gas- and particle-phase product distribution). As shown in the α-pinene + OH reaction scheme (Fig. 1) acetone
        is formed when the monocyclic alkoxy radical decomposes via $O_2$ addition. For limonene photooxidation (Fig. 7e), the

concentration of acetone is lower than for α-pinene and more of the C8 to C10 gaseous species are further oxidized and partitioned
        into the particle phase (Fig. S9c). This resulted in a final distribution of 50 % gas-phase organic products, 20 % particle-phase
        organic products, and 30 % $CO+CO_2$. The simulated acetone yields are qualitatively consistent with experimental data that have
        shown yields of acetone from α-pinene photooxidation (Lee et al., 2006b; Wisthaler et al., 2001) can be up to four orders of
        magnitude higher than from limonene photooxidation (Lee et al., 2006b; Reissell et al., 1999).

For the α-pinene ozonolysis system (Fig. 7d), at the end of the simulation 88 % of the carbon is gas-phase organic products,
        7 % particle-phase organic products, and 5 % $CO+CO_2$. For limonene ozonolysis (Fig. 7f), 50 % of the carbon fraction is gas-
        phase organics, 43 % particle-phase organics, and 7 % $CO+CO_2$. The higher particle-phase fraction for limonene ozonolysis is a
        result of the C8 and C10 organic products of limonene being more highly functionalized and thus partitioned to the particle phase
        (Figs. S9d and S13); whereas the C8 and C10 organic products of α-pinene are more volatile and partitioned to the gas phase (Figs.

6a and S9b).

### 3.2 Controlled reactivity simulations

        The GECKO-A simulations captured trends (e.g., SOA yields and major products) observed in chamber studies (section
        3.1) for α-pinene and limonene (two common terpene model surrogates). Therefore, the GECKO-A model was used to perform a
        detailed study of SOA formation from camphene under idealized atmospheric ("controlled reactivity") conditions, which were

compared with analogous simulations for α-pinene and limonene.

#### 3.2.1 Gas-phase chemistry

        Time-dependent mixing ratios of $HO_2$, OH, and $NO_3$ are shown in Fig. 8 for the controlled reactivity simulations
        performed at 0.1 ppb of $HC_o$ (camphene, α-pinene, or limonene) and 10 µg m$^{-3}$ of organic seed. The $O_3$ and total $NO_x$ levels were
        fixed so that the oxidant (OH, $O_3$, and $NO_3$) levels would remain stable during the simulations. The time profiles of $HO_2$, OH, and

$NO_3$ are independent of the precursor, confirming that the gas-phase oxidant levels are controlled by the added ethane and
        formaldehyde. This allows for a comparative assessment of the monoterpenes. The reaction rate of camphene with $O_3$ is extremely
        slow (two and three orders of magnitude lower than the rate constants for α-pinene+$O_3$ and limonene+$O_3$ respectively (Atkinson
        and Arey, 2003a)); thus camphene predominantly reacts with OH in the simulations, while α-pinene and limonene react with $O_3$
        and OH (see Fig. S26 for relative fractions).

#### 3.2.2 Simulated SOA formation

        Figure 9 illustrates the simulated SOA yields as a function of atmospheric aging time (Fig. 9a) and the SOA yield as
        function of reacted HC concentration (Fig. 9b) for the controlled reactivity simulations. The atmospheric aging time, τ is defined
        as:

$$\tau = \frac{1}{[OH]_{atm}} \int_0^t [OH]_{sim} dt \tag{1}$$

where $[OH]_{atm}$ is the atmospheric OH concentration ($2 \times 10^6$ molecule cm$^{-3}$ was assumed) and $[OH]_{sim}$ is the simulated OH
        concentration. Camphene was predicted to form more SOA (0.26 µg m$^{-3}$) than α-pinene (0.14 µg m$^{-3}$) but less than limonene (0.42





µg m$^{-3}$) after 14.5 hours of aging time (Fig. 9a). The simulation results in Fig. 9b show that camphene, which reacts predominantly with OH (Fig. S26), forms low volatility products (more SOA at lower ΔHC) at the start of the reaction than α-pinene and limonene. However, after the precursor is completely consumed, the SOA yield of limonene exceeds that of camphene due to a higher fraction

of limonene oxidation products reacting further to form extremely low volatility products. As previously reported (Lee et al., 2006b), and as simulated herein, limonene has the highest SOA yield among well studied monoterpenes. However, the final SOA yield of camphene was relatively high, approximately twice that of α-pinene.

### 3.2.3 Gas- and particle-phase product distribution

Figure 10 shows the product distribution in the gas- and particle-phases after 72 hours (equivalent to 14.5 hours of

atmospheric OH aging time) for the controlled reactivity simulations. While thousands of secondary species are formed during the oxidation of a given monoterpene, only species that contribute ≥ 0.01 % of the total gas- or particle-phase mass were included in Fig. 10. Also, all C1 species, as well as seven of the C2 gas-phase products (whose concentrations were largely a direct result of ethane chemistry) were omitted from Fig. 10. For camphene (Fig. 10a), the particle phase is largely dominated by C10 species with 3 to 5 functional groups, followed by highly functionalized C7 species (typically with 4 to 5 functional groups). Similarly,

for limonene (Fig. 10b), the particle phase is dominated by C10 species with 4 to 5 functional groups, followed by C7 to C9 species with 4 to 5 functional groups. However, for α-pinene (Fig. 10c), there is a broad distribution of C8 to C10 products (with 3 to 4 functional groups) contributing to the particle phase. Generally, the volatility of particle-phase products from camphene and limonene was lower than from α-pinene. As shown in Fig. 10a, a large fraction of gas-phase products from camphene, as compared to limonene, is composed of C9 and C10 products whose volatility was not low enough to partition to the particle phase. This

further explains the SOA yields shown in Fig. 9b where limonene SOA yield exceeded camphene SOA yield at the end of the reaction.

Figure 11 shows the final mass percentages of α-pinene, camphene, and limonene particle-phase oxidation products grouped into three volatility categories. The volatility categories were assigned based on the calculated mass saturation concentrations (C*) of the simulated products. Log C* values in the range of < -3.5, -3.5 to -0.5, and -0.5 to 2.5 were assigned

respectively as extremely low-volatility, low-volatility, and semi-volatile organic compounds (ELVOCs, LVOCs, and SVOCs) (Chuang and Donahue, 2016; Zhang et al., 2015). Limonene, which had the highest simulated SOA yield among the three studied monoterpenes, was largely LVOCs (59 %), followed by ELVOCs (24 %) and then SVOCs (17 %). Camphene SOA was also largely LVOCs (67 %), followed by SVOCs (28 %), and then a significantly lower fraction of ELVOCs (4 %) than limonene. In contrast, α-pinene SOA was dominated by SVOCs (50 %), followed by LVOCs (48 %), and then ELVOCs (2 %). For α-pinene

and camphene, intermediate-volatility organic compounds (IVOCs) were less than 1 % of the SOA mass. For experimental studies of α-pinene ozonolysis, Zhang et al. (2015) reported a fractional contribution of ~68 % SVOCs to final SOA mass, which is similar to the contribution predicted using GECKO-A.

### 3.2.4 Using α-pinene limonene as a surrogate for camphene

For the controlled reactivity simulations, the final SOA mass and yield of camphene (0.26 µg m$^{-3}$, 0.46) were between the

final SOA mass and yield of α-pinene (0.14 µg m$^{-3}$, 0.25) and limonene (0.42 µg m$^{-3}$, 0.74). This suggests that camphene could potentially be represented in models as a 50/50 mixture of α-pinene + limonene. To test this, a controlled reactivity simulation was run with 50 ppt α-pinene + 50 ppt limonene; simulation results were then compared with the simulation results for 0.1 ppb of camphene. Figure 12a shows that while the slopes of the SOA yield curves differ over the course of the reaction, the SOA masses (0.26 µg m$^{-3}$ for 50 % α-pinene + 50 % limonene and 0.26 µg m$^{-3}$ for camphene) and yields (0.46 for 50 % α-pinene + 50 %





limonene and 0.47 for camphene) were approximately equal at the end of the simulation. However, the end of simulation particle-phase volatility distributions (Fig. 12b) are notably different. The 50 % α-pinene + 50 % limonene simulation had a significantly higher fraction (25 %) of ELVOCs, influenced by the low volatility limonene products, than the camphene simulation (4 %). These results suggest that while the final SOA mass and yield of the 50/50 α-pinene + limonene mixture were representative of camphene, the properties (e.g., volatility) of the particle-phase products were not. The volatility distributions will influence the formation of

SOA at the lowest mass loadings and will also influence changes in SOA mass as a function of dilution, with the surrogate mixture (50 % α-pinene + 50 % limonene) producing less volatile SOA than predicted for camphene. Thus, the extent to which camphene can be represented by α-pinene + limonene will depend on the application. To improve the representation of camphene, a second simulation was run with 50 ppt α-pinene + 50 ppt limonene, where the rate constants of α-pinene and limonene were replaced with the rate constants of camphene during the chemical mechanism generation. However, the representation of camphene SOA by the

50/50 α-pinene + limonene mixture did not improve (resulted in higher final SOA yield of 0.51) when the rate constants of α-pinene and limonene were replaced with those of camphene (Fig. 12a). Also, representing camphene by the limonene mechanism with camphene rate constants did not improve the representation of camphene SOA (see Fig. S29). This illustrates the importance of both the reaction rates and structure on SOA formation from monoterpenes.

   To demonstrate the potential impact of including a parameterized representation of SOA formation by camphene in air

quality models, SOA mass and yields were predicted for three wildland fire fuels based on the measured monoterpene distributions in Hatch et al. (2015) for black spruce, and Hatch et al. (2019) for Douglas fir and lodgepole pine. The top five monoterpenes by emissions factor (mass of compound emitted/mass fuel burned) represent ~70-80 % of the total monoterpene emission factor (EF) for each of these fuels. These top five monoterpenes were used to represent SOA formation for each fuel by normalizing the EF for each fuel; assigning α-pinene as the model surrogate for all measured compounds except limonene, including camphene; and

then reassigning camphene as 50 % α-pinene and 50 % limonene. SOA mass concentrations and yields were predicted assuming a background PM level of 50 µg m$^{-3}$ and ΔHC = 10 ppb and using published two-product SOA parameters based on Griffin et al. (1999) (Table S3) and volatility basis set (VBS) parameters (low NO$_x$, dry) based on Pathak et al. (2007b) (for α-pinene) and Zhang et al. (2006) (for limonene) (Table S4). The two model parameterizations were used to represent a range of potential outcomes. The SOA yields using the two-product parameters were lower than predicted here for α-pinene (~0.1), but similar for

camphene (~0.6); using the VBS parameters, the yields were similar for α-pinene (~0.2) but higher than predicted here for camphene (~0.9). The total OA mass loadings in the parameterized SOA calculations were a factor of 3-6 higher than in the GECKO-A controlled reactivity simulations, which is consistent with the higher SOA yield for camphene predicted using the VBS parameters. The results of the SOA calculations are summarized in Table 4. For lodgepole pine, there is no change in SOA mass, because camphene is not one of the top five monoterpenes by EF. However, for fuels in which camphene contributed significantly

to the measured monoterpene EF, SOA mass increased by 43-50 % for black spruce and by 56-108 % for Douglas fir.

## 4 Conclusions

   While camphene is a ubiquitous monoterpene, measured in significant quantities from both biogenic and pyrogenic sources, little is known about SOA formation from camphene and there are no published parameterizations to represent camphene SOA in air quality models. GECKO-A simulations suggest that the initial organic oxidation products of camphene are of low

volatility and can condense at low OA mass loadings; lower than predicted for α-pinene and limonene. Predicted final SOA yields for camphene in the controlled reactivity simulations (~45 %) were in between those predicted for α-pinene (25 %) and limonene (~75 %), suggesting that SOA formation from camphene can be represented in air quality models assuming a 50/50 (α-



pinene/limonene) surrogate mixture. Calculations based on measured monoterpene distributions for three wildland fire fuels illustrate that accounting for camphene, in this case using the surrogate mixture and published SOA parameterizations for α-pinene and limonene, increased predicted SOA mass by 43-108 %. This demonstrates the potential impact of representing SOA formation from camphene in air quality models, and the need for an appropriate parameterization. The surrogate mixture appears to represent the SOA mass and yield of camphene well, but not necessarily the volatility distribution of the products. Further modeling and/or experimental studies are needed to develop and test a suitable SOA parameterization for representing camphene in air quality models.

**Author contribution**

IA performed the model simulations and led analysis and visualization efforts. KB conceptualized, administered, and supervised the project. BA and MC developed the software and methodology, including the model, and supported research design and interpretation of the results. IA prepared the manuscript with review and editing contributions from KB, BA, and MC.

**Competing interests**

The authors declare that they have no conflict of interest.

**Acknowledgements**

The authors would like to thank Richard Valorso for the training given in the use of GECKO-A modeling tools. IA and KB acknowledge support from the National Oceanic and Atmospheric Administration (NOAA) grant AC4 NA16OAR4310103 and the National Science Foundation grant AGS-1753364.

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

**Table 1: Initial conditions for α-pinene and limonene chamber reactivity simulations.**

| Abbreviation | Description | HC (ppb) | NO (ppb) | HONO (ppb) | $O_3$ (ppb) | Organic seed ($\mu$g m$^{-3}$) |
|---|---|---|---|---|---|---|
| P_LHC | Photooxidation Lower Hydrocarbon | 50 | 110 | 10 | | 1 |
| P_HHC | Photooxidation Higher Hydrocarbon | 150 | 110 | 10 | | 1 |
| DO_LHC | Dark Ozonolysis Lower Hydrocarbon | 50 | 16 | | 500 | 1 |
| DO_HHC | Dark Ozonolysis Higher Hydrocarbon | 150 | 16 | | 500 | 1 |

**Table 2: Initial conditions for camphene, α-pinene, and limonene controlled reactivity simulations. The levels of $O_3$ and $NO_x$ were fixed during these simulations.**

| Abbreviation | Description | HC (ppb) | NO (ppb) | $O_3$ (ppb) | $C_2H_6$ (ppm) | $CH_2O$ (ppb) | Organic seed ($\mu$g m$^{-3}$) |
|---|---|---|---|---|---|---|---|
| CR | Controlled Reactivity | 0.1 | 1 | 30 | 10 | 50 | 10 |






**Table 3: Calculated average mass-weighted O/C ratio and OH exposure at the end of the α-pinene and limonene photooxidation and ozonolysis simulations.**

|  | Average O/C | OH exposure (molec cm$^{-3}$ s) |
|---|---|---|
| α-Pinene photooxidation | 0.93 | $6.7 \times 10^{10}$ |
| α-Pinene ozonolysis | 0.64 | $1.5 \times 10^{10}$ |
| Limonene photooxidation | 0.96 | $9.1 \times 10^{10}$ |
| Limonene ozonolysis | 0.68 | $1.7 \times 10^{10}$ |

**Table 4: SOA yield and mass predicted using 2-product and VBS parameters for top five monoterpenes by emission factor (EF) from black spruce, Douglas fir, and lodgepole pine. For each fire fuel, the monoterpenes were represented using two different surrogate assignments. In Assignment 1, α-pinene is used to represent all monoterpenes except limonene. In Assignment 2, camphene is represented as 50 % α-pinene and 50 % limonene. There is no Assignment 2 for lodgepole pine, because camphene is not one of the top five monoterpenes by EF. The percentage (%) increase in SOA was calculated based on the difference between the total SOA of assignments 1 and 2. The EFs of assignments 1 and 2, the 2-product SOA parameters, and the VBS parameters are provided in Tables S2, S3, and S4, respectively.**

|  |  | Yield$_{\alpha pin}$ | Yield$_{lim}$ | SOA$_{\alpha pin}$ (µg m$^{-3}$) | SOA$_{lim}$ (µg m$^{-3}$) | SOA$_{total}$ (µg m$^{-3}$) | % increase in SOA |
|---|---|---|---|---|---|---|---|
| | | | | *Black Spruce* | | | |
| 2-Product | Assignment 1 | 0.099 | 0.6 | 4.5 | 6.4 | 10.9 | 50 % |
| | Assignment 2 | 0.103 | 0.6 | 3.6 | 12.8 | 16.4 | |
| VBS | Assignment 1 | 0.194 | 0.93 | 8.8 | 9.9 | 18.7 | 43 % |
| | Assignment 2 | 0.202 | 0.93 | 7 | 19.7 | 26.7 | |
| | | | | *Douglas Fir* | | | |
| 2-Product | Assignment 1 | 0.098 | 0.6 | 4.5 | 6.1 | 10.6 | 108 % |
| | Assignment 2 | 0.108 | 0.6 | 2.5 | 19.6 | 22.1 | |
| VBS | Assignment 1 | 0.194 | 0.93 | 8.9 | 9.4 | 18.3 | 56 % |
| | Assignment 2 | 0.203 | 0.93 | 6.6 | 21.9 | 28.5 | |
| | | | | *Lodgepole Pine* | | | |
| 2-Product | Assignment 1 | 0.097 | 0.6 | 4.7 | 4.6 | 9.3 | |
| VBS | Assignment 1 | 0.192 | 0.93 | 9.3 | 7.1 | 16.4 | |





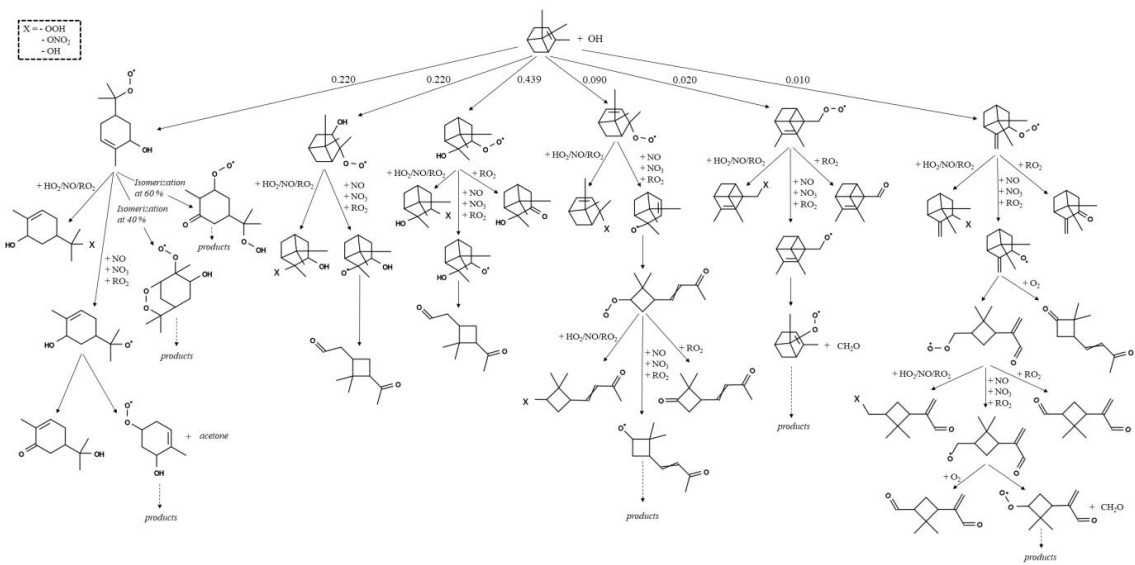

**Figure 1: Initial oxidation pathways of α-pinene oxidation with OH as represented in GECKO-A.**

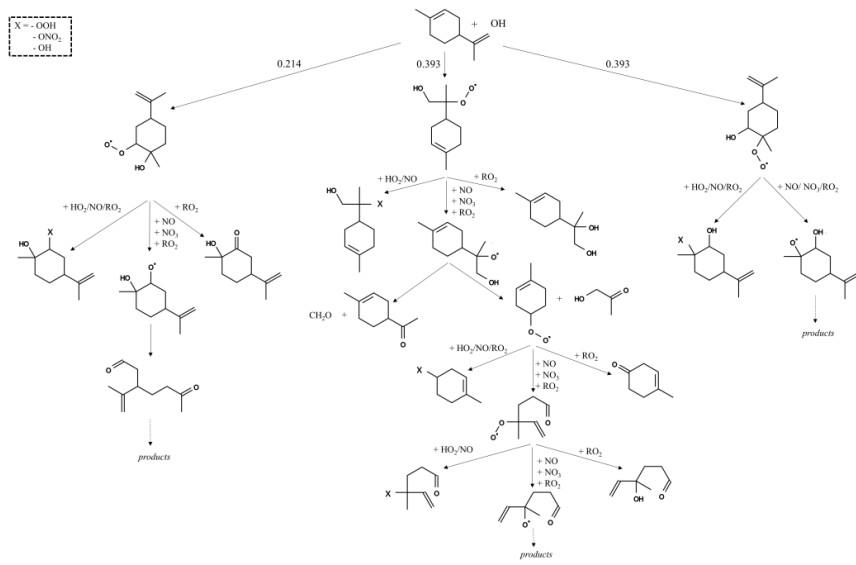

**Figure 2: Initial oxidation pathways of limonene oxidation with OH as represented in GECKO-A.**




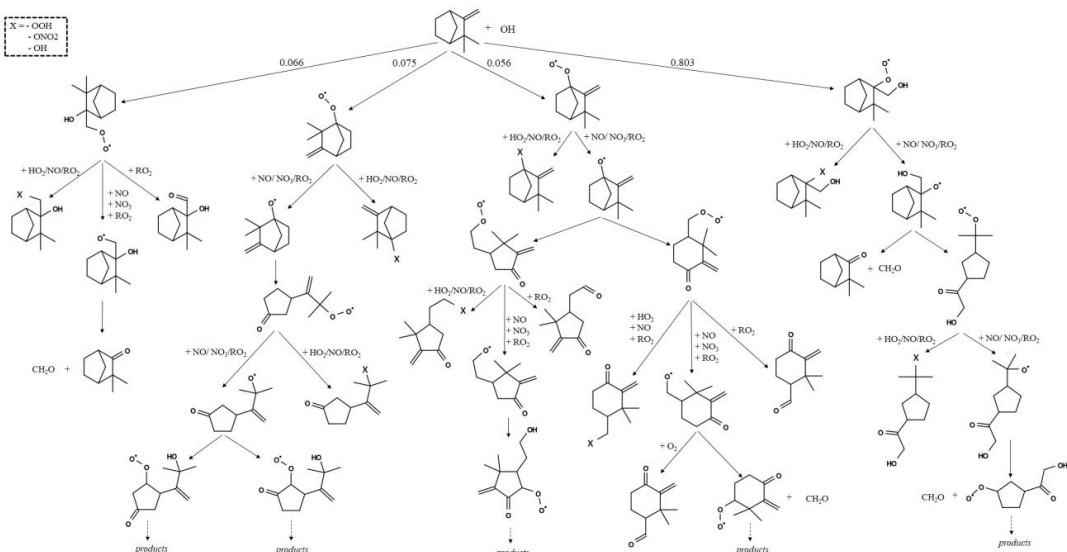

**Figure 3: Initial oxidation pathways of camphene oxidation with OH as represented in GECKO-A.**






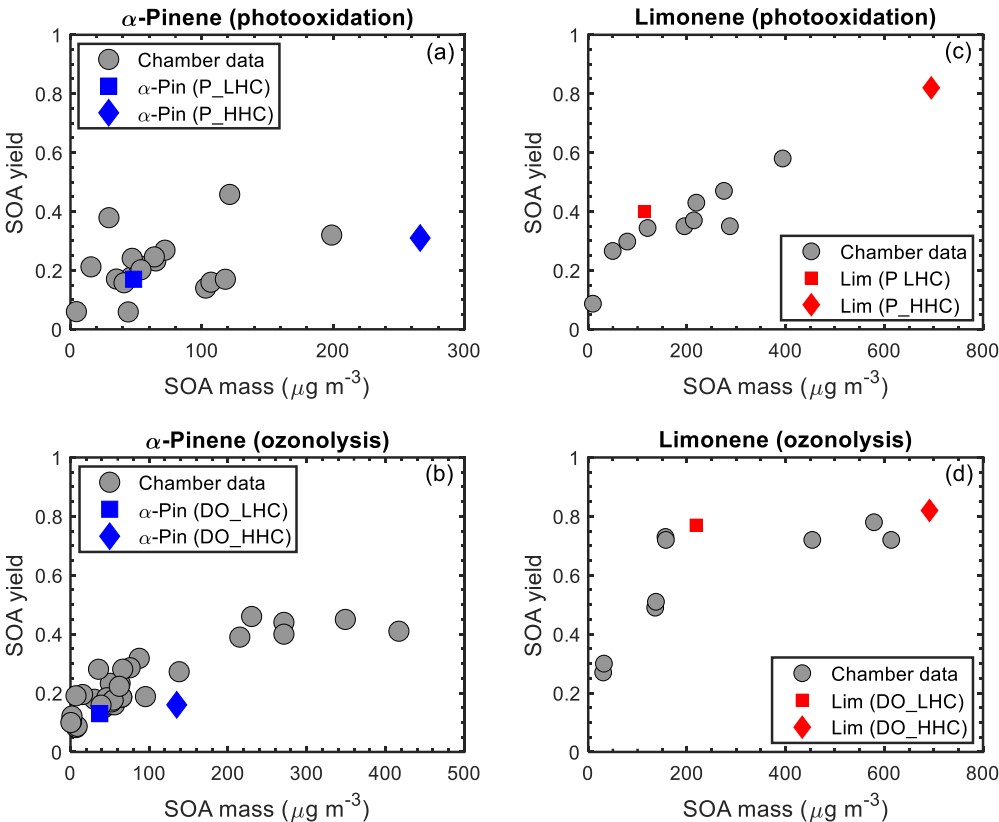

**Figure 4: GECKO-A SOA yields are represented by blue (α-pinene) and red (limonene) markers; chamber SOA yields are represented by grey markers. The initial hydrocarbon mixing ratios are differentiated by shape; squares represent the simulation using the lower hydrocarbon (LHC) mixing ratio and diamonds the simulation using the higher hydrocarbon (HHC) mixing ratio.**






propan-2-one

Molecular Formula: C$_3$H$_6$O

2-(3-acetyl-2,2-dimethylcyclobutyl)
-1-(nitroperoxy)ethan-1-one

Molecular Formula: C$_{10}$H$_{15}$NO$_6$

1-(nitroperoxy)ethan-1-one

Molecular Formula: C$_2$H$_3$NO$_5$

(3-acetyl-2,2-dimethylcyclobutyl)
methyl nitrate

Molecular Formula: C$_9$H$_{15}$NO$_4$

(3-acetyl-2,2-dimethylcyclobutyl)
acetaldehyde

Molecular Formula: C$_{10}$H$_{16}$O$_2$

(3-acetyl-2,2-dimethylcyclobutyl)
acetic acid

Molecular Formula: C$_{10}$H$_{16}$O$_3$

3-acetyl-2,2-dimethylcyclobutyl
nitrate

Molecular Formula: C$_8$H$_{13}$NO$_4$

3-(nitroperoxy)-3-oxopropanal

Molecular Formula: C$_3$H$_3$NO$_6$

2-methyl-3-[2-(nitroperoxy)-2-
oxoethyl]cyclobutan-1-one

Molecular Formula: C$_7$H$_9$NO$_6$

(2,2-dimethyl-3-oxocyclobutyl)
acetaldehyde

Molecular Formula: C$_8$H$_{12}$O$_2$

**Figure 5a: Simulated top 10 gas-phase products from α-pinene photooxidation (P_LHC).**


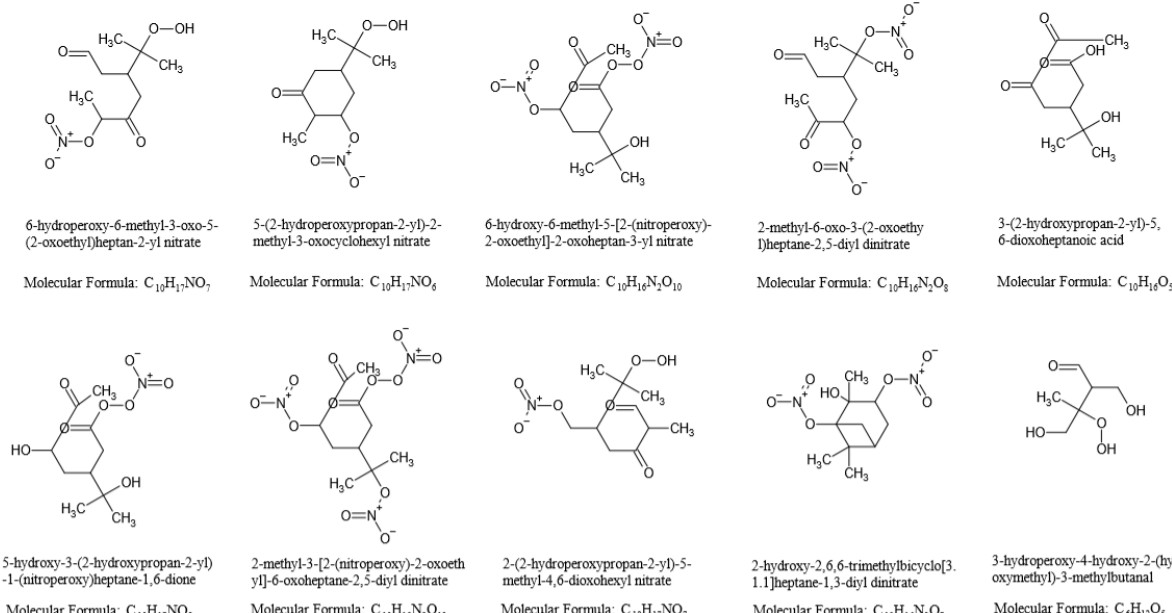

6-hydroperoxy-6-methyl-3-oxo-5-
(2-oxoethyl)heptan-2-yl nitrate

Molecular Formula: C$_{10}$H$_{17}$NO$_7$

5-(2-hydroperoxypropan-2-yl)-2-
methyl-3-oxocyclohexyl nitrate

Molecular Formula: C$_{10}$H$_{17}$NO$_6$

6-hydroxy-6-methyl-5-[2-(nitroperoxy)-
2-oxoethyl]-2-oxoheptan-3-yl nitrate

Molecular Formula: C$_{10}$H$_{16}$N$_2$O$_{10}$

2-methyl-6-oxo-3-(2-oxoethy
l)heptane-2,5-diyl dinitrate

Molecular Formula: C$_{10}$H$_{16}$N$_2$O$_8$

3-(2-hydroxypropan-2-yl)-5,
6-dioxoheptanoic acid

Molecular Formula: C$_{10}$H$_{16}$O$_5$

5-hydroxy-3-(2-hydroxypropan-2-yl)
-1-(nitroperoxy)heptane-1,6-dione

Molecular Formula: C$_{10}$H$_{17}$NO$_8$

2-methyl-3-[2-(nitroperoxy)-2-oxoeth
yl]-6-oxoheptane-2,5-diyl dinitrate

Molecular Formula: C$_{10}$H$_{15}$N$_3$O$_{12}$

2-(2-hydroperoxypropan-2-yl)-5-
methyl-4,6-dioxohexyl nitrate

Molecular Formula: C$_{10}$H$_{17}$NO$_7$

2-hydroxy-2,6,6-trimethylbicyclo[3.
1.1]heptane-1,3-diyl dinitrate

Molecular Formula: C$_{10}$H$_{16}$N$_2$O$_7$

3-hydroperoxy-4-hydroxy-2-(hydr
oxymethyl)-3-methylbutanal

Molecular Formula: C$_6$H$_{12}$O$_5$

**Figure 5b: Simulated top 10 particle-phase products from α-pinene photooxidation (P_LHC).**






**Figure 6a: Top 10 gas-phase products from α-pinene dark ozonolysis (DO_LHC) simulations.**


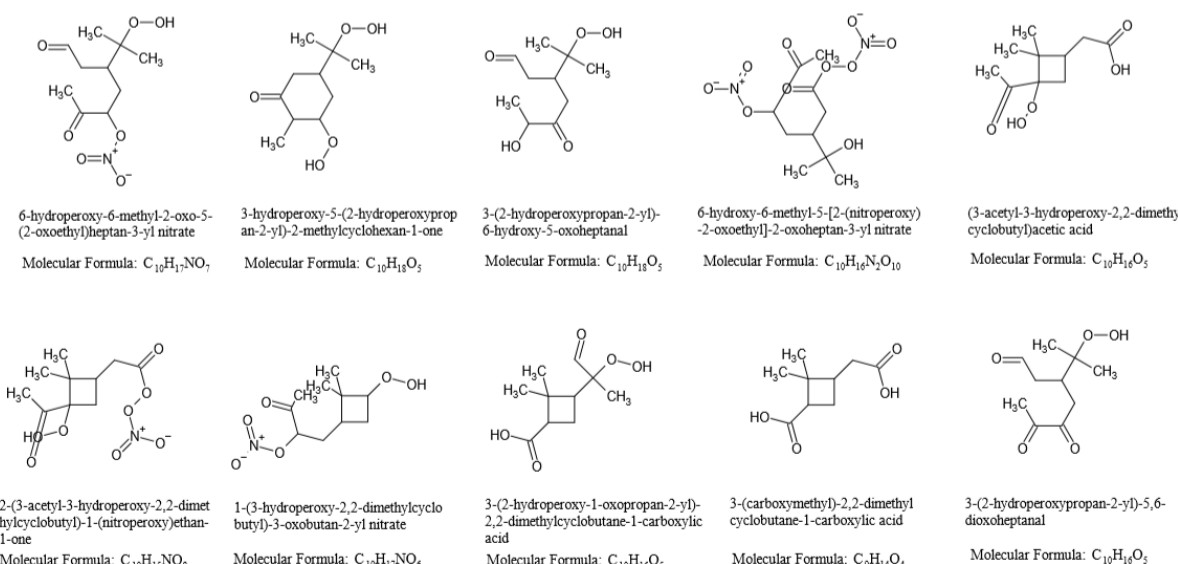

**Figure 6b: Top 10 particle-phase products from α-pinene dark ozonolysis (DO_LHC) simulations.**





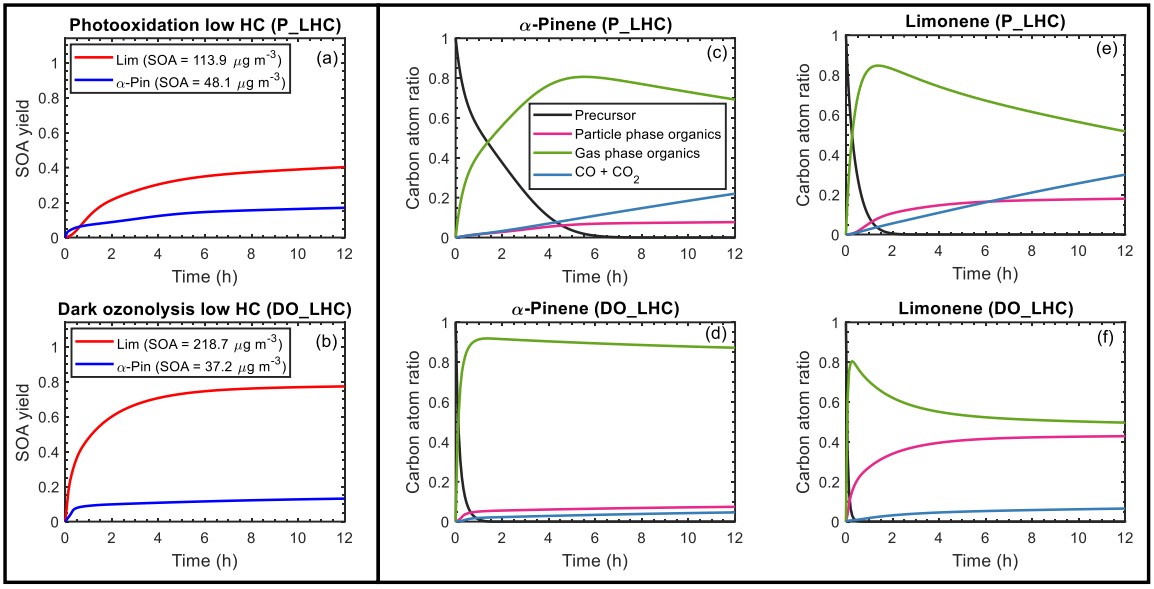

**Figure 7: Simulated SOA yield (a and b) and carbon budget (c to f) as a function of time for α-pinene and limonene during photooxidation (a, c, e) and dark ozonolysis (b, d, f). The SOA yield curve for α-pinene is represented by a blue line; limonene is represented by a red line. For the carbon budget plots, the mixing ratios of the precursor (black line), particle-phase organics (magenta line), gas-phase organics (green line), and CO+CO₂ (blue line) are expressed as carbon atom ratios (ppbC/initial precursor in ppbC). The results shown are for the low hydrocarbon mixing ratio (50 ppb) simulations.**

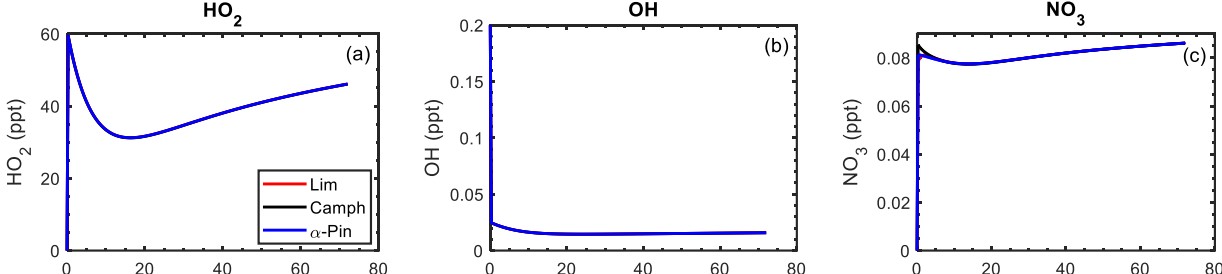

**Figure 8: Mixing ratios of HO₂, OH, and NO₃ as function of time for limonene (red line), camphene (black line), and α-pinene (blue line) during the controlled reactivity simulations with 0.1 ppb of HC₀ and 10 μg m⁻³ of organic seed. By design, the profiles of the mixing ratios for each precursor overlap except for at the very beginning of the NO₃ profile.**





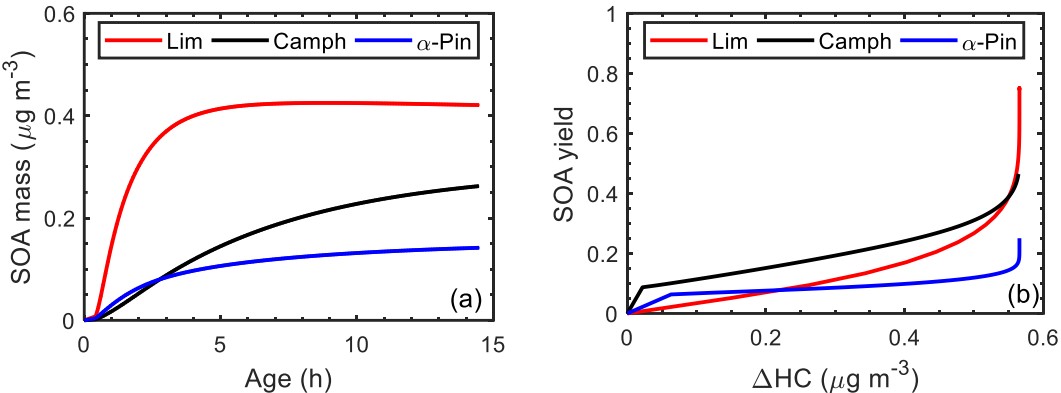

**Figure 9: (a) Simulated SOA mass as a function of atmospheric aging time (reaction with OH) and (b) simulated SOA yield as a function of reacted hydrocarbon concentration (ΔHC) during controlled reactivity simulation at 0.1 ppb HCₒ with 10**
**μg m⁻³ seed for limonene (red line), camphene (black line), and α-pinene (blue line).**





**Figure 10: Number of functional groups associated with gas- and particle-phase species as a function of carbon number. Results are shown for camphene, α-pinene, and limonene after 72 hours of oxidation under controlled reactivity condition. The markers are sized by the ratio of their mixing ratio (in ppbC) to the initial mixing ratio of the precursor (in ppbC). The colors of the markers are scaled by volatility (represented by saturation concentration, C*).**





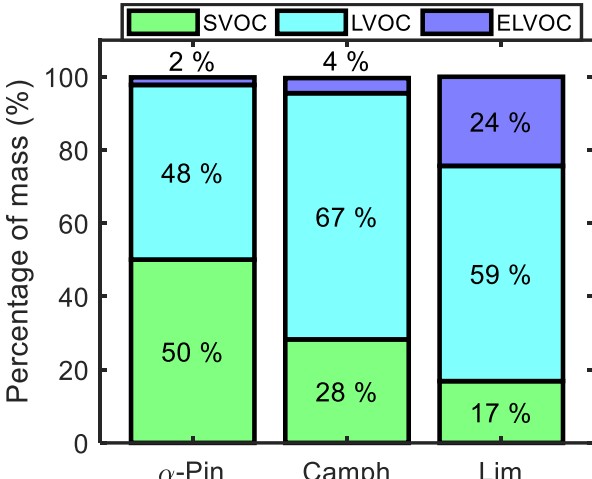


**Figure 11: Mass percentage of four volatility categories in the particle phase at the end of the controlled reactivity simulations for α-pinene, camphene, and limonene.**


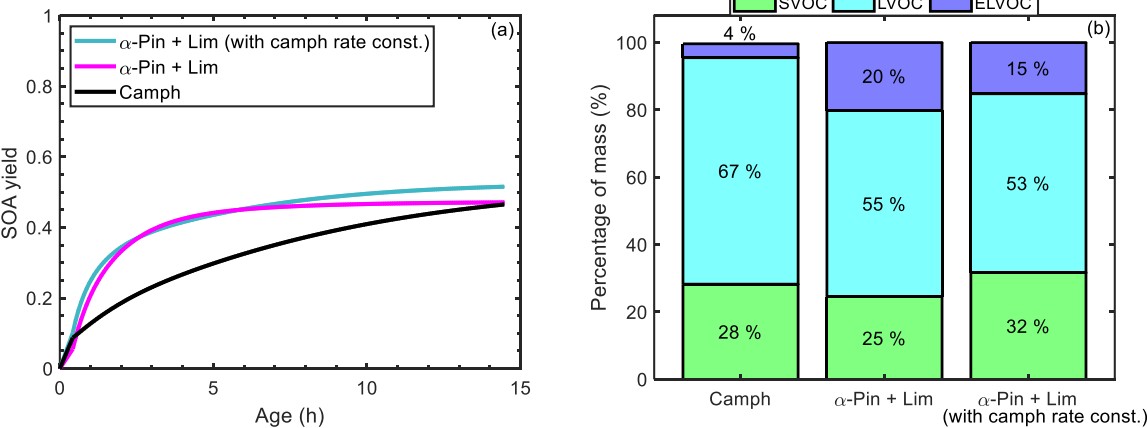

**Figure 12: (a) Simulated SOA yield as a function of atmospheric aging time for camphene (black line), 50 % α-pinene + 50 % limonene (magenta line), and 50 % α-pinene + 50 % limonene where the rate constants of α-pinene and limonene were replaced with the rate constants of camphene (green line); and (b) mass percentage of four volatility categories in the particle phase at the end of the controlled reactivity simulations for camphene, 50 % α-pinene + 50 % limonene, and limonene, and 50 % α-pinene + 50 % limonene where the rate constants of α-pinene and limonene were replaced with the rate constants of camphene.**
