# Peer review of "Using GECKO-A to derive mechanistic understanding of SOA formation from the ubiquitous but understudied camphene"

_Atmospheric Chemistry and Physics, 2020_

## Referee Comment (RC1) · Anonymous Referee #1 · 21 Oct 2020

The authors present a unique study in which a near-explicit chemical mechanism generator (GECKO-A) is used to consider the oxidation and SOA formation potential of camphene, a monoterpene of importance as a biomass burning emission, but that has received only limited attention in chamber studies. Comparisons are made with two well-studied species, a-pinene and limonene. The study is, in my opinion, very well conceived, thoroughly conducted and well written. I have some questions and comments below (mostly minor) that the author should consider prior to final publication in ACP.

It looks to me as though some of the initial steps in the OH/a-pinene and maybe

OH/limonene are specified (to match chamber data?), whereas camphene chemistry is presumably all derived from 'free-running' GECKO. If true, could this bias the results in any way?

Page 3, line 107: Can anything more be said to justify the selection of six generations of oxidation - e.g. were any of the previous studies alluded to conducted on monoterpenes?

Bottom right of Figure 3, the co-product should be acetone instead of formaldehyde.

Page 4, line 128: The peroxy radicals formed seem mostly to still contain the double-ring structure of a-pinene (not just the 4-membered ring?).

Page 4, line 135: I think the compound formed is camphenilone?

Page 5, line 151: Did you mean acetylperoxy radical, rather than acetaldehyde?

Page 5, line 154: camphenilone spelled incorrectly.

Page 5, line 155-157: You might mention here that an OH radical is also generated.

Page 6, line 220: What precursor(s) were studied by Shilling et al.?

Page 7: line 228: This is maybe not be a fully addressable question at this point, but would the presence of OH scavengers likely have a major effect on the results?

Page 7, line 232: It might be worth emphasizing that these top-ten lists presumably evolve with reaction time.

Some of the structures in Figures 5 and 6 look a little bit strange (with overlapping or extended bonds). Can these be cleaned up a bit?

Figure 9b: Converting the x-axis units to ppb would be useful, I think, since these are the units used throughout the text.

Page 9, near line 305: Could the high SOA yield from limonene (and its by-products) be in part driven by its shorter lifetime (hence giving more time in the simulations for

oxidation of products?).

---

## Referee Comment (RC2) · Anonymous Referee #2 · 18 Nov 2020

Afreh et al. present findings from a box modeling study performed on camphene, a monoterpene emitted by vegetation and biomass burning. After validating their model on more well-studied monoterpenes: a-pinene and limonene, they find that camphene has a high potential to form secondary organic aerosol (SOA), which is likely to alter SOA estimates from real-world biomass burning that emits a distribution of monoterpenes.

Although global emissions of monoterpenes are dominated by only a few species (a-pinene, b-pinene, limonene), the remaining monoterpenes that include camphene could play an important role locally and regionally. Yet their oxidation chemistry and

ability to contribute to SOA production remains uncertain. Hence this study is well-motivated. In addition, it provides a pathway for a similar analysis to be done on both anthropogenic and biogenic volatile organic compounds (VOCs) for which little experimental data are available to determine their atmospheric impacts. However, for a few key reasons, I found the methods, results, and interpretation to be either incomplete or not accurately described. I believe that while this work has the potential, I cannot recommend publication in Atmospheric Chemistry and Physics at this time. See detailed comments below.

Major comments:

1. Autooxidation and oligomerization: There is strong evidence that many monoterpenes participate in autooxidation reactions to form highly oxygenated organic molecules (HOM) and participate in condensed phase reactions to form high molecular weight oligomers (Bianchi et al., Chemical Reviews, 2019). Although the authors mention that this is not considered in this work, *I believe this is a major shortcoming that has the potential to confound the findings for camphene and its comparison with the findings for a-pinene and limonene*. At the very least, this needs to be dealt with in a simplistic manner. For instance, molar yields leading to HOM could be tied to experimental observations (Ehn et al., Nature, 2014; Jokinen et al., PNAS, 2015) and oligomer formation could be informed by reaction rates in the literature (e.g., Ziemann and Atkinson, Chem. Soc. Review, 2012; Roldin et al., ACP, 2014).

2. Section 2.1: Many of the monoterpene oxidation products are likely to be in the semi-volatile range, which co-exist in the gas and particle phases. Point '(2)' makes it seem like the SOA production was limited to only include low-volatility products that are below a certain vapor pressure (on a related note, it would be nice to specify the vapor pressure in C* units of $\mu$g/m3). I don't think that is the case but this needs to be explicitly mentioned. Related to this, how are C* values for all oxidation products calculated? Depending on the method used, how does one think about the uncertainty in the methods used to calculate the C*? The fractional contribution of semi-volatile

organic compounds to the product distribution is likely to affect the SOA mass yield significantly.

3. Comparison with prior chamber data: While the qualitative comparison offered in Figure 4 with prior chamber data is useful, there are many more reasons than those stated in the first paragraph in Section 3.1.1 (lines 190-209) that could explain the differences between the different studies and those studies and this work. A few of them being: differences in vapor and size-dependent particle wall losses, seeded versus unseeded, differences in total oxidant exposure, NO versus NO2 availability, lights used and photolysis rates, relative humidity and availability of aerosol water, and aerosol acidity. Trying to describe differences in prior data exclusively based on their NOx and OA loading differences as well as commenting that model is able to capture the general behavior, is too simplistic and ignores many of the nuances linked to chamber experiments. This section needs to be significantly expanded if the goal is to demonstrate that the model can capture previous observations of SOA mass yields.

4. Application to wildfire SOA (lines 354-370): I am generally in favor of this analysis but do not agree with the approach used here and the context in which they are presented. How consistent are the VBS parameters between the studies chosen (i.e., Griffin, Pathak, Zhang) and the other studies alluded to earlier in Table S1? *Why were the MCM results from the previous sections not used directly to perform this analysis? My sense is that these could easily be used to develop yield estimates for a-pinene, limonene, and camphene.* Further, the analysis does not seem to accurately represent camphene but only compares estimates using the two different approximations (i.e., camphene = a-pinene or camphene = a-pinene+limonene). The projected enhancements for the different fuels are hence unconstrained. Finally, the work should be presented in the context of other studies that have attempted to model SOA formation from biomass burning emissions and the relative importance of monoterpenes or lack of it to other SOA precursors (e.g., phenols, furans, aromatics). See, for example, the work of Bruns et al. (Sci. Rep., 2016), Ahern et al., (JGR, 2019), and Akherati et

al., (ES&T, 2020).

5. One key aspect that this study fails to highlight – but one that is quite powerful - is the general approach to thinking about SOA formation from unexplored VOCs. I would recommend that a revision of this manuscript highlight this aspect. Particularly, since camphene, regardless of how it is dealt with, may not be important enough to meaningfully affect the total SOA burden from vegetation or biomass burning.

Minor comments (not exhaustive):

1. Line 53: Reaction rates or reaction rate constants? Having very different reaction rate constants doesn't necessarily mean that the atmospheric lifetimes could vary by the same amount.

2. Introduction: Well-cited section but could perhaps also be motivated by how these monoterpene emissions and their composition could change with future temperatures.

3. Line 86: 'compared'.

4. Section 2.2: How are the generalized reaction schemes for monoterpenes determined?

5. Line 186: Why was 50 degrees selected?

6. For the results discussed in Section 3.1, can time series plots for SOA mass, O:C, and major components be included in the main text or SI for the simulations?

7. Table 3: This only shows modeled O:C values. Can a comparison similar to Figure 4 be also be performed for SOA O:C, in addition to or instead of the comparison to literature values in the second paragraph in Section 3.1.1? On a related note, HOM formation will tend to further increase the modeled O:C and further separate the modeled O:C from the literature data. Thoughts on why this might be? Is this related to not having enough semi-volatile material with low O:C to 'dilute' the bulk O:C?

8. Line 225: It would seem like a lot of $H_2O$ would need to be lost (and lot of oligomers

produced in the process) to explain the differences in the modeled and measured O:C. Can this calculation be done to test plausibility?

9. Line 227: It should have been relatively easy to zero out the OH in the model simulations to appropriately represent the chamber experiments where an OH scavenger was used.

10. Section 3.1.2: The volatility distribution of a-pinene SOA has been well studied (e.g., Saha and Grieshop, ES&T, 2016; Yli-Juuti et al., GRL, 2017) and to some degree its composition too (e.g., Sato et al., ACP, 2018; D'Ambro et al., ES&T, 2018). While a qualitative comparison has been done here, can this be done more quantitatively against literature data? For instance, could the modeled SOA products be lumped by volatility and compared against volatility distributions constructed from analyzing dilution, thermodenuder, and speciation data? Can the modeled composition be compared directly to measured data at the species level? This would help improve confidence in the model predictions.

11. Line 254: Should it be 'macroscopic'?

12. Line 263: Do you see an effect of photolysis on SOA mass yields? This recent study might be of interest to examine for consistency with this work: https://doi.org/10.1021/acs.est.9b07051

13. Line 288: Were these values picked to replicate atmospheric conditions? If yes, this should be stated. Was there a reason the model was not used to explore dependence on T, RH, NOx, etc?

14. Section 3.2: If SOA production was assumed to be instantaneous, would the kinetics under low condensational sink conditions, alter the model predictions significantly under atmospheric conditions?

15. Section 3.2.3: How would HOM production and differences between the three VOCs affect the modeled particle-phase product distribution and its volatility? This

should have a reasonable impact on the results shown in Figure 11.

16. Line 375: 'lower than' what?

17. Line 380: Needs to be made clear that this is only for the SOA contribution from monoterpenes, which could be quite small, depending on the fuel type.

---

## Author Comment (AC1) · 4 Feb 2021

1.1: The authors present a unique study in which a near-explicit chemical mechanism generator (GECKO-A) is used to consider the oxidation and SOA formation potential of camphene, a monoterpene of importance as a biomass burning emission, but that has received only limited attention in chamber studies. Comparisons are made with two well-studied species, a-pinene and limonene. The study is, in my opinion, very well conceived, thoroughly conducted and well written. I have some questions and comments below (mostly minor) that the author should consider prior to final publication in ACP.

[Figure]

1.2: It looks to me as though some of the initial steps in the OH/a-pinene and maybe OH/limonene are specified (to match chamber data?), whereas camphene chemistry is presumably all derived from 'free-running' GECKO. If true, could this bias the results in any way?

Response: The initial rate constants (kOH, kO3, and kNO3) of $\alpha$-pinene, camphene, and limonene were specified based on data from Atkinson and Arey, 2003. The initial branching ratios of $\alpha$-pinene + OH were based on data from Peeters et al., 2001. Reaction products and rate constants of subsequent reaction steps of $\alpha$-pinene, and the full camphene and limonene mechanisms were based on SARs. Therefore, we do not expect bias in the results due to differences in prescribed vs. free-running mechanism generation. To improve clarity, the following text has been added in lines 128-134: The monoterpene reaction schemes are generated by GECKO-A using established protocols, as described in Aumont et al. (2005). First, the mechanism generator analyzes the structure of the compound to determine the reactive sites and the plausible reaction pathways. Reaction products and initial branching ratios are based on experimental data when available. Otherwise, the reaction products and rate constants are estimated based on structure-activity relationships (SARs). The initial reaction rate constants of the monoterpenes with OH, O3, and NO3 were based on data from Atkinson and Arey (2003a). For $\alpha$-pinene + OH, the initial branching ratios are based on data from Peeters et al. (2001). For subsequent reactions steps with $\alpha$-pinene + OH and for the limonene and camphene mechanisms, reaction products and branching ratios are based on SARs.

1.3: Page 3, line 107: Can anything more be said to justify the selection of six generations of oxidation - e.g. were any of the previous studies alluded to conducted on monoterpenes?

Response: The selection of six generations of oxidation was based on previous studies on n-alkanes (Aumont et al., 2012). This has been clarified on line 112. In addition, we performed sensitivity studies early on that confirmed no significant change in the

evolution of the gas and particle phase from 6 to 7 generations. We have added the following text on lines 112-113: "sensitivity studies performed in this work for $\alpha$-pinene oxidation, where increasing the number of generations beyond six did not result in significant changes in the evolution of the gas and particle phases;"

1.4: Bottom right of Figure 3, the co-product should be acetone instead of formaldehyde

Response: Corrected.

1.5: Page 4, line 127: The peroxy radicals formed seem mostly to still contain the double ring structure of a-pinene (not just the 4-membered ring?).

Response: Agreed. Line 142 has been edited to clarify that it is the stable products, and not the peroxy radicals, which are predominantly four-membered ring species.

1.6: Page 4, line 135: I think the compound formed is camphenilone?

Response: Corrected.

1.7: Page 5, line 151: Did you mean acetylperoxy radical, rather than acetaldehyde?

Response: Yes. "Acetaldehyde" has been removed; the term "peroxy radicals" represents all peroxy radicals formed, including the acetylperoxy radical (lines 166-167). "...the alkoxy radicals either react with O2 or decompose to form formaldehyde and peroxy radicals."

1.8: Page 5, line 154: camphenilone spelled incorrectly

Response: Corrected.

1.9: Page 5, line 155-157: You might mention here that an OH radical is also generated.

Response: We did not include inorganic species in the mechanism figures. The following sentence has been added to the text in lines 137-138: For figure clarity, inorganic species formed (including OH) are not shown.; and to the figure description (Figures 1,

2, 3, S1, S2, and S3) to clarify.

1.10: Page 6, line 220: What precursor(s) were studied by Shilling et al.?

Response: The sentence has been revised by adding "for $\alpha$-pinene".

1.11: Page 7: line 228: This is maybe not be a fully addressable question at this point, but would the presence of OH scavengers likely have a major effect on the results?

Response: Sensitivity simulations were performed using CO as an OH scavenger. The differences in calculated O/C ratios for both alpha-pinene and limonene were small, and insufficient to explain the differences between the model simulations and observations. Though this is not an exhaustive study of the potential effects of an OH scrubber on composition, given the results of the sensitivity simulations we have removed the suggestion that this may explain model-measurement discrepancy.

1.12: Page 7, line 232: It might be worth emphasizing that these top-ten lists presumably evolve with reaction time.

Response: The sentence has been revised to note that these are the top 10 products at the end of the simulation.

1.13: Some of the structures in Figures 5 and 6 look a little bit strange (with overlapping or extended bonds). Can these be cleaned up a bit?

Response: Yes. Corrected. Thank you for the suggestion.

1.14: Figure 9b: Converting the x-axis units to ppb would be useful, I think, since these are the units used throughout the text.

Response: The units on the x-axis in Figure 9b have been changed to ppb.

1.15: Page 9, near line 305: Could the high SOA yield from limonene (and its by-products) be in part driven by its shorter lifetime (hence giving more time in the simulations for C2 oxidation of products?).

Response: Yes. The lower volatility products formed from limonene oxidation are expected to be in part a result of the shorter lifetime of limonene and its oxidation products. This behavior is also expected to be influenced by (i) the relative contribution of gaseous oxidants (here mainly OH, but also O3 for a-pinene and limonene) and (ii) the structure of the formed alkoxy radicals, and thus their chemical evolution by fragmentation/functionalization (e.g. Camredon et al., 2007). The following sentence has been added in lines 324 – 326 "The shorter lifetime and the chemical structure, including presence of two double bonds, contribute to the relatively high SOA yield of limonene."

Please also note the supplement to this comment:
https://acp.copernicus.org/preprints/acp-2020-829/acp-2020-829-AC1-supplement.pdf

[Figure]

[Figure]

**Fig. 1.** Figure 3: Initial oxidation pathways of camphene oxidation with OH as represented in GECKO-A (inorganic products are not shown).

[Figure]

**Fig. 2.** Fig9b:simulated SOA yield as a function of reacted hydrocarbon concentration (ïĄĎHC) during controlled reactivity simulation

**Supplement:**

| | Reviewer 1 | Responses to Reviewers: We thank the reviewers for their detailed and insightful comments and questions. They have led to an improved manuscript. Line numbers refer to the revised manuscript, *tracked changes version*. |
|---|---|---|
| | **Summary** | |
| 1.1 | The authors present a unique study in which a near-explicit chemical mechanism generator (GECKO-A) is used to consider the oxidation and SOA formation potential of camphene, a monoterpene of importance as a biomass burning emission, but that has received only limited attention in chamber studies. Comparisons are made with two well-studied species, a-pinene and limonene. The study is, in my opinion, very well conceived, thoroughly conducted and well written. I have some questions and comments below (mostly minor) that the author should consider prior to final publication in ACP. | No response required. |
| | **Comments** | |
| 1.2 | It looks to me as though some of the initial steps in the OH/a-pinene and maybe OH/limonene are specified (to match chamber data?), whereas camphene chemistry is presumably all derived from 'free-running' GECKO. If true, could this bias the results in any way? | The initial rate constants ($k_{OH}$, $k_{O3}$, and $k_{NO3}$) of α-pinene, camphene, and limonene were specified based on data from Atkinson and Arey, 2003. The initial branching ratios of α-pinene + OH were based on data from Peeters et al., 2001. Reaction products and rate constants of subsequent reaction steps of α-pinene, and the full camphene and limonene mechanisms were based on SARs. Therefore, we do not expect bias in the results due to differences in prescribed vs. free-running mechanism generation. To improve clarity, the following text has been added in lines 128-134: *The monoterpene reaction schemes are generated by GECKO-A using established protocols, as described in Aumont et al. (2005). First, the mechanism generator analyzes the structure of the compound to determine the reactive sites and the plausible reaction pathways. Reaction products and initial branching ratios are based on experimental data when available. Otherwise, the reaction products and rate constants are estimated based on structure-activity relationships (SARs). The initial reaction rate constants of the monoterpenes with OH, $O_3$, and $NO_3$ were based on data from Atkinson and Arey (2003a). For α-pinene + OH, the initial branching ratios are based on data from Peeters et al. (2001). For subsequent reactions steps with α-pinene + OH and for the limonene and camphene mechanisms, reaction products and branching ratios are based on SARs.* |
| 1.3 | Page 3, line 107: Can anything more be said to justify the selection of six generations of oxidation - e.g. were any of the previous studies alluded to conducted on monoterpenes? | The selection of six generations of oxidation was based on previous studies on n-alkanes (Aumont et al., 2012). This has been clarified on line 112. In addition, we performed sensitivity studies early on that confirmed no significant change in the evolution of the gas and particle phase from 6 to 7 generations. |

| | | We have added the following text on lines 112-113: *"sensitivity studies performed in this work for α-pinene oxidation, where increasing the number of generations beyond six did not result in significant changes in the evolution of the gas and particle phases;"* |
|---|---|---|
| 1.4 | Bottom right of Figure 3, the co-product should be acetone instead of formaldehyde | Corrected. |
| 1.5 | Page 4, line 127: The peroxy radicals formed seem mostly to still contain the double ring structure of a-pinene (not just the 4-membered ring?). | Agreed. Line 142 has been edited to clarify that it is the stable products, and not the peroxy radicals, which are predominantly four-membered ring species. |
| 1.6 | Page 4, line 135: I think the compound formed is camphenilone? | Corrected. |
| 1.7 | Page 5, line 151: Did you mean acetylperoxy radical, rather than acetaldehyde? | Yes. "Acetaldehyde" has been removed; the term "peroxy radicals" represents all peroxy radicals formed, including the acetylperoxy radical (lines 166-167). *"...the alkoxy radicals either react with $O_2$ or decompose to form formaldehyde and peroxy radicals."* |
| 1.8 | Page 5, line 154: camphenilone spelled incorrectly | Corrected. |
| 1.9 | Page 5, line 155-157: You might mention here that an OH radical is also generated. | We did not include inorganic species in the mechanism figures. The following sentence has been added to the text in lines 137-138: *For figure clarity, inorganic species formed (including OH) are not shown.*; and to the figure description (Figures 1, 2, 3, S1, S2, and S3) to clarify. |
| 1.10 | Page 6, line 220: What precursor(s) were studied by Shilling et al.? | The sentence has been revised by adding "for α-pinene". |
| 1.11 | Page 7: line 228: This is maybe not be a fully addressable question at this point, but would the presence of OH scavengers likely have a major effect on the results? | Sensitivity simulations were performed using CO as an OH scavenger. The differences in calculated O/C ratios for both alpha-pinene and limonene were small, and insufficient to explain the differences between the model simulations and observations. Though this is not an exhaustive study of the potential effects of an OH scrubber on composition, given the results of the sensitivity simulations we have removed the suggestion that this may explain model-measurement discrepancy. |
| 1.12 | Page 7, line 232: It might be worth emphasizing that these top-ten lists presumably evolve with reaction time. | The sentence has been revised to note that these are the top 10 products at the end of the simulation. |
| 1.13 | Some of the structures in Figures 5 and 6 look a little bit strange (with overlapping or extended bonds). Can these be cleaned up a bit? | Yes. Corrected. Thank you for the suggestion. |

| | | |
|---|---|---|
| 1.14 | Figure 9b: Converting the x-axis units to ppb would be useful, I think, since these are the units used throughout the text. | The units on the x-axis in Figure 9b have been changed to ppb. |
| 1.15 | Page 9, near line 305: Could the high SOA yield from limonene (and its by-products) be in part driven by its shorter lifetime (hence giving more time in the simulations for C2 oxidation of products?). | Yes. The lower volatility products formed from limonene oxidation are expected to be in part a result of the shorter lifetime of limonene and its oxidation products. This behavior is also expected to be influenced by (i) the relative contribution of gaseous oxidants (here mainly OH, but also $O_3$ for a-pinene and limonene) and (ii) the structure of the formed alkoxy radicals, and thus their chemical evolution by fragmentation/functionalization (e.g. Camredon et al., 2007). The following sentence has been added in lines 324 – 326 "*The shorter lifetime and the chemical structure, including presence of two double bonds, contribute to the relatively high SOA yield of limonene.*" |

---

## Author Response (AR1)

**Responses to Reviewers (acp-2020-829)**

**Manuscript: "Using GECKO-A to derive mechanistic understanding of SOA formation from the ubiquitous but understudied camphene"**

**Reviewer Comment 1**

**Summary**

**Comment 1.1:** The authors present a unique study in which a near-explicit chemical mechanism generator (GECKO-A) is used to consider the oxidation and SOA formation potential of camphene, a monoterpene of importance as a biomass burning emission, but that has received only limited attention in chamber studies. Comparisons are made with two well-studied species, α-pinene and limonene. The study is, in my opinion, very well conceived, thoroughly conducted and well written. I have some questions and comments below (mostly minor) that the author should consider prior to final publication in ACP.

**Response 1.1:** No response required.

**Comments**

**Comment 1.2:** It looks to me as though some of the initial steps in the OH/ α-pinene and maybe OH/limonene are specified (to match chamber data?), whereas camphene chemistry is presumably all derived from 'free-running' GECKO. If true, could this bias the results in any way?

**Response 1.2:** The initial rate constants ($k_{OH}$, $k_{O3}$, and $k_{NO3}$) of α-pinene, camphene, and limonene were specified based on data from Atkinson and Arey, 2003. The initial branching ratios of α-pinene + OH were based on data from Peeters et al., 2001. Reaction products and rate constants of subsequent reaction steps of α-pinene, and the full camphene and limonene mechanisms were based on SARs. Therefore, we do not expect bias in the results due to differences in prescribed vs. free-running mechanism generation. To improve clarity, the following text has been added in lines 128-134:

*"The monoterpene reaction schemes are generated by GECKO-A using established protocols, as described in Aumont et al. (2005). First, the mechanism generator analyzes the structure of the compound to determine the reactive sites and the plausible reaction pathways. Reaction products and initial branching ratios are based on experimental data when available. Otherwise, the reaction products and rate constants are estimated based on structure-activity relationships (SARs). The initial reaction rate constants of the monoterpenes with OH, $O_3$, and $NO_3$ were based on data from Atkinson and Arey (2003a). For α-pinene + OH, the initial branching ratios are based on data from Peeters et al. (2001). For subsequent reactions steps with α-pinene + OH and for the limonene and camphene mechanisms, reaction products and branching ratios are based on SARs."*

**Comment 1.3:** Page 3, line 107: Can anything more be said to justify the selection of six generations of oxidation - e.g. were any of the previous studies alluded to conducted on monoterpenes?

**Response 1.3:** The selection of six generations of oxidation was based on previous studies on n-alkanes (Aumont et al., 2012). This has been clarified on line 112. In addition, we performed sensitivity studies early on that confirmed no significant change in the evolution of the gas and particle phase from 6 to 7 generations. We have added the following text on lines 112-113:

*"sensitivity studies performed in this work for α-pinene oxidation, where increasing the number of generations beyond six did not result in significant changes in the evolution of the gas and particle phases;"*

**Comment 1.4:** Bottom right of Figure 3, the co-product should be acetone instead of formaldehyde

**Response 1.4:** Corrected.

**Comment 1.5:** Page 4, line 127: The peroxy radicals formed seem mostly to still contain the double ring structure of α-pinene (not just the 4-membered ring?).

**Response 1.5:** Agreed. Line 143 has been edited to clarify that it is the stable products, and not the peroxy radicals, which are predominantly four-membered ring species:

*"...which then react with NO, $RO_2$ or $HO_2$ to form stable products, many with a four-membered ring,..."*

**Comment 1.6:** Page 4, line 135: I think the compound formed is camphenilone?

**Response 1.6:** Corrected.

**Comment 1.7:** Page 5, line 151: Did you mean acetylperoxy radical, rather than acetaldehyde?

**Response 1.7:** Yes. "Acetaldehyde" has been removed; the term "peroxy radicals" represents all peroxy radicals formed, including the acetylperoxy radical (lines 166-167):

*". . .the alkoxy radicals either react with $O_2$ or decompose to form formaldehyde and peroxy radicals."*

**Comment 1.8:** Page 5, line 154: camphenilone spelled incorrectly

**Response 1.8:** Corrected.

**Comment 1.9:** Page 5, line 155-157: You might mention here that an OH radical is also generated.

**Response 1.9:** We did not include inorganic species in the mechanism figures. The following sentence has been added to the text in lines 137-138 and to the figure description (Figures 1, 2, 3, S1, S2, S3, S4, S5, and S6) to clarify:

*"For figure clarity, inorganic species formed (including OH) are not shown."*

**Comment 1.10:** Page 6, line 220: What precursor(s) were studied by Shilling et al.?

**Response 1.10:** The sentence has been revised by adding "for α-pinene" in line 238:

*"... for α-pinene ..."*

**Comment 1.11:** Page 7: line 228: This is maybe not be a fully addressable question at this point, but would the presence of OH scavengers likely have a major effect on the results?

**Response 1.11:** Sensitivity simulations were performed using CO as an OH scavenger. The differences in calculated O/C ratios for both alpha-pinene and limonene were small, and insufficient to explain the differences between the model simulations and observations. Though this is not an exhaustive study of the potential effects of an OH scrubber on composition, given the results of the sensitivity simulations we have removed the suggestion that this may explain model-measurement discrepancy.

The sentence below has been removed (lines 245-247):

*"Additionally, for dark ozonolysis, the OH scavengers typically present in chamber experiments (e.g., Kourtchev et al., 2015; Shilling et al., 2009) were absent from the simulations."*

**Comment 1.12:** Page 7, line 232: It might be worth emphasizing that these top-ten lists presumably evolve with reaction time.

**Response 1.12:** The sentence has been revised to note that these are the top 10 products at the end of the simulation (lines: 256-258):

*"Figures 5a and 5b show the chemical structures and molecular formulae of the top 10 products by mass in the gas and particle phases at the end of the α-pinene photooxidation simulation."*

**Comment 1.13:** Some of the structures in Figures 5 and 6 look a little bit strange (with overlapping or extended bonds). Can these be cleaned up a bit?

**Response 1.13:** Yes. Corrected. Thank you for the suggestion.

**Comment 1.14:** Figure 9b: Converting the x-axis units to ppb would be useful, I think, since these are the units used throughout the text.

**Response 1.14:** The units on the x-axis in Figure 9b have been changed to ppb.

**Comment 1.15:** Page 9, near line 305: Could the high SOA yield from limonene (and its byproducts) be in part driven by its shorter lifetime (hence giving more time in the simulations for C2 oxidation of products?).

**Response 1.15:** Yes. The lower volatility products formed from limonene oxidation are expected to be in part a result of the shorter lifetime of limonene and its oxidation products. This behavior is also expected to be influenced by (i) the relative contribution of gaseous oxidants (here mainly OH, but also $O_3$ for α-pinene and limonene) and (ii) the structure of the formed alkoxy radicals, and thus their chemical evolution by fragmentation/functionalization (e.g. Camredon et al., 2007). The following sentence has been added in lines 324 – 326:

*"The shorter lifetime and the chemical structure, including presence of two double bonds, contribute to the relatively high SOA yield of limonene."*

**Reviewer Comment 2**

**Summary**

**Comment 2.1:** Afreh et al. present findings from a box modeling study performed on camphene, a monoterpene emitted by vegetation and biomass burning. After validating their model on more well-studied monoterpenes: α-pinene and limonene, they find that camphene has a high potential to form secondary organic aerosol (SOA), which is likely to alter SOA estimates from real-world biomass burning that emits a distribution of monoterpenes. Although global emissions of monoterpenes are dominated by only a few species (α-pinene, b-pinene, limonene), the remaining monoterpenes that include camphene could play an important role locally and regionally. Yet their oxidation chemistry and ability to contribute to SOA production remains uncertain. Hence this study is well motivated. In addition, it provides a pathway for a similar analysis to be done on both anthropogenic and biogenic volatile organic compounds (VOCs) for which little experimental data are available to determine their atmospheric impacts. However, for a few key reasons, I found the methods, results, and interpretation to be either incomplete or not accurately described. I believe that while this work has the potential, I cannot recommend publication in Atmospheric Chemistry and Physics at this time. See detailed comments below.

**Response 2.1:** No response required.

**Major Comments**

**Comments 2.2:** Autooxidation and oligomerization: There is strong evidence that many monoterpenes participate in autooxidation reactions to form highly oxygenated organic molecules (HOM) and participate in condensed phase reactions to form high molecular weight oligomers (Bianchi et al., Chemical Reviews, 2019). Although the authors mention that this is not considered in this work, *I believe this is a major shortcoming that has the potential to confound the findings for camphene and its comparison with the findings for α-pinene and limonene*. At the very least, this needs to be dealt with in a simplistic manner. For instance, molar yields leading to HOM could be tied to experimental observations (Ehn et al., Nature, 2014; Jokinen et al., PNAS, 2015) and oligomer formation could be informed by reaction rates in the literature (e.g., Ziemann and Atkinson, Chem. Soc. Review, 2012; Roldin et al., ACP, 2014).

**Response 2.2:** We agree with the reviewer that HOM formation and subsequent dimerization in the gas-phase is likely to be an important contributor to SOA formation under certain conditions, as has been demonstrated in laboratory, field, and modeling studies. We note HOM formation+dimerization has been particularly studied in the context of new particle formation, and at low SOA mass loadings, both of which are not relevant here. Efforts are underway to develop a SAR to predict HOM formation that is suitable to run in GECKO-A. This is not trivial to achieve, mainly as the H-shift $RO_2$ reactions would lead to a non-manageable number of species and reactions if treated explicitly. Reduction protocols are currently under development to consider these classes of reactions in GECKO-A. A full consideration of HOM formation/dimerization is however not possible at this time. To that end, the following has been added in lines 104-109:

*"Autoxidation, including the formation of highly oxygenated molecules (HOM) in the gas phase (Bianchi et al., 2019; Ehn et al., 2014), is not currently represented in GECKOA. A SAR to predict the rate coefficients of peroxy radical (RO₂) H migration reactions (H-shifts) that lead to the formation of HOM, was recently published by Vereecken and Nozière (2020). The straight implementation of this SAR into GECKO-A would lead to a non-manageable number of species and reactions. Reduction protocols are thus currently under development to consider the autoxidation reactions in subsequent model versions."*

Also, the following revision has been made in lines 125-126:

*"Condensed-phase reactions are not currently represented in GECKO-A."*

Nonetheless, based on the reviewer's suggestion, we have done some calculations regarding $RO_2$ lifetimes with respect to different terminal reactions and some estimations of the potential importance of HOM formation + dimerization for SOA formation. Guided by Bianchi et al., 2019 and references therein, $RO_2$ lifetimes with respect to NO, $HO_2$, and $RO_2$ were calculated for all of the simulation conditions. As has been previously shown and summarized in Bianchi et al. 2019, the lifetime of $RO_2$ with NO is too short ($< 7$ s in our simulations) under photooxidation conditions with NOx for HOM formation to play a significant role in gas-phase chemistry. Whereas under dark ozonolysis conditions, the lifetime of $RO_2$ is long enough such that HOM formation may play a role in gas-phase chemistry and SOA formation. Calculations of gas-phase HOM and HOM-dimer yields as a function of time are qualitatively consistent with Ehn et al., 2014, that at the beginning of the simulations (for the fastest assumed rate constant), where $RO_2$ concentrations are highest and SOA mass concentrations are lowest, HOM dimers likely contribute to SOA mass and may result in an increase in the predicted SOA yield. However, as the SOA mass increases in time, the contribution of such dimers is outweighed by the contribution of more abundant SVOCs.

That $RO_2$ dimerization may contribute to SOA formation at the beginning of the dark ozonolysis (DO) chamber condition simulations does not change the key discussion points or conclusion of the manuscript. Of more relevance are the controlled reactivity simulations, which were designed such that $RO_2$ reacts ∼equally with $HO_2$ and NO. The lifetime of $RO_2$ is $< 60$ s in such simulations, and thus, we do not expect $RO_2$ HOM formation and dimerization to significantly affect the gas-phase chemistry or SOA formation. Further, given the range of HOM yields as a function of reaction conditions and parent compound structure, and the lack of available information for camphene, efforts to calculate HOM formation, dimerization, and SOA formation in these simulations would be overly speculative and would not likely contribute any concrete findings on the importance of such chemistry, particularly for camphene.

Regarding heterogeneous/particle-phase accretion reactions, in McVay et al., 2016 particle-phase dimerization was considered for three reactions using two different rate constants, and the results were compared with chamber studies. For both high and low UV conditions the predicted SOA yield increased; while this improved measurement-model agreement for the low UV experiments, it resulted in a significant over prediction in the high UV experiments. For the chamber simulation conditions in this work, the time-dependent product distributions (by functional group) were similar for alpha-pinene and limonene, suggesting that the inclusion of accretion reactions may

also increase the simulated alpha-pinene and limonene SOA yields in this work. Similarly to McVay et al., this would likely improve measurement-model agreement in some cases (e.g., alpha-pinene ozonlysis) but not others. Though as noted here and in the manuscript, the goal was not to reproduce any one chamber experiment and a comprehensive suite of sensitivity studies for potentially relevant processes is out of the scope of this work. For the controlled reactivity conditions, no published study exists from which we may draw insight. As with the chamber simulations, the monomer building blocks are present in the particle phase, but simply calculating an accretion yield for these monomers based on published accretion reaction rates is not reasonable because the subsequent gas-particle partitioning can't be considered as it was in McVay et al. 2016. The large number of gas- and particle-phase species ($9.3 \times 104 - 2 \times 105$) precludes running simple "offline" or out of GECKO-A box models. Modifying the GECKO-A source code to treat accretion reactions is beyond the scope of this work. We acknowledge that it may be important, and will continue to work on a feasible way to consider such reactions in GECKO-A.

In acknowledgement of these important comments, the following sentence has been added to the conclusions (lines 407-408):

*"The SOA mass, yields and product volatility distributions can also be influenced by gas-phase HOM formation and dimerization, and accretion reactions, which were not considered here."*

**Comment 2.3:** Section 2.1: Many of the monoterpene oxidation products are likely to be in the semi-volatile range, which co-exist in the gas and particle phases. Point '(2)' makes it seem like the SOA production was limited to only include low-volatility products that are below a certain vapor pressure (on a related note, it would be nice to specify the vapor pressure in C* units of μg/m3). I don't think that is the case but this needs to be explicitly mentioned. Related to this, how are C* values for all oxidation products calculated? Depending on the method used, how does one think about the uncertainty in the methods used to calculate the C*? The fractional contribution of semi-volatile organic compounds to the product distribution is likely to affect the SOA mass yield significantly.

**Response 2.3:** Because we have specific product structures available, vapor pressures are calculated for each compound using the vapor pressure estimation method of Nannoolal, and it is the saturation vapor pressures (and not saturation vapor concentrations, C*) that are used to calculate the partitioning between the gas and condensed phase. A number of studies have been published comparing vapor pressure estimation methods, and the effects of uncertainties on partitioning predictions. For example, Valorso et al. (2011) compared several estimation methods of saturation vapor pressures for secondary organic compounds formed during α-pinene oxidation. As is noted by the reviewer, partitioning of the semi-volatile compounds may be particularly affected by such uncertainties. For α-pinene oxidation, it has been shown that a better measurement-model agreement is obtained when the Nannoolal method is used, especially under high NOx conditions (e.g. Valorso et al. (2011)). This method is thus recommended for saturation vapor pressure estimation in GECKO-A. Based on the reviewer comments, the following additions or revisions have been made.

Lines 124-125:

*". . . which performs relatively well compared to other estimation methods when used to simulate SOA formation during α-pinene oxidation experiments (Valorso et al., 2011)."*

Lines 114-116 have been revised as follows:

*". . .species with saturated vapor pressure below 10-13 atm (equivalent to C\* of 1.02 × 10-3 µg m-3 for species with a mean molecular weight of 250 g mol-1) were considered non-volatile and therefore treated as end products during gas-phase mechanism generation;"*

Also, a sentence describing the C\* calculation has been added in lines 344-345:

*"C\* was calculated based on the equilibrium absorption coefficient equation, as defined by Odum et al. (1996) and Pankow (1994)."*

**Comment 2.4:** Comparison with prior chamber data: While the qualitative comparison offered in Figure 4 with prior chamber data is useful, there are many more reasons than those stated in the first paragraph in Section 3.1.1 (lines 190-209) that could explain the differences between the different studies and those studies and this work. A few of them being: differences in vapor and size-dependent particle wall losses, seeded versus unseeded, differences in total oxidant exposure, NO versus $NO_2$ availability, lights used and photolysis rates, relative humidity and availability of aerosol water, and aerosol acidity. Trying to describe differences in prior data exclusively based on their NOx and OA loading differences as well as commenting that model is able to capture the general behavior, is too simplistic and ignores many of the nuances linked to chamber experiments. This section needs to be significantly expanded if the goal is to demonstrate that the model can capture previous observations of SOA mass yields.

**Response 2.4:** We agree with the reviewer and recognize that a large number of factors influence SOA mass concentrations and yields observed in chamber studies. One of the reasons we compiled data from 12 different studies was to account for a range of wall loss effects (as influenced by initial conditions, including seed concentrations, and chamber size), temperature and relative humidity (though the vast majority of studies are conducted under dry (< 10% RH) conditions), light intensity, etc. We note in the manuscript that the scatter in the experimental yield data shown in Fig. 4a is due to differences in experimental conditions. We do not attempt to reproduce any individual experiment (and note this in line 183: *"Here, no attempt is made to strictly reproduce the conditions of a given chamber experiment."*).

Rather we are evaluating whether the model captures general trends in SOA yields and products, to the extent the latter can be evaluated. We believe that the analysis presented supports the conclusion that the model represents the chamber data (at macroscopic and molecular levels) sufficiently well that it is reasonable to use GECKO-A to probe the potential for camphene to form SOA, relative to alpha-pinene and limonene.

**Comment 2.5:** Application to wildfire SOA (lines 354-370): I am generally in favor of this analysis but do not agree with the approach used here and the context in which they are presented. How consistent are the VBS parameters between the studies chosen (i.e., Griffin, Pathak, Zhang)

and the other studies alluded to earlier in Table S1? *Why were the MCM results from the previous sections not used directly to perform this analysis? My sense is that these could easily be used to develop yield estimates for α-pinene, limonene, and camphene.* Further, the analysis does not seem to accurately represent camphene but only compares estimates using the two different approximations (i.e., camphene = α-pinene or camphene = α-pinene+limonene). The projected enhancements for the different fuels are hence unconstrained. Finally, the work should be presented in the context of other studies that have attempted to model SOA formation from biomass burning emissions and the relative importance of monoterpenes or lack of it to other SOA precursors (e.g., phenols, furans, aromatics). See, for example, the work of Bruns et al. (Sci. Rep., 2016), Ahern et al., (JGR, 2019), and Akherati et al., (ES&T, 2020).

**Response 2.5:** The results of the GECKO-A modeling simulations are used to guide this analysis, from the perspective that they indicated that camphene may be represented by a mix of alpha-pinene and limonene. The benefit of using these compounds to represent camphene, is that experimentally-based SOA parameterizations are available for such compounds, and are not available for camphene.

In lines 358-359, the following has been added:

*"...for which SOA parameterizations currently are available."*

The GECKO-A modeling simulations do not cover sufficient parameter space to develop a robust SOA parameterization. There are certainly differences in SOA modeling parameters due to differences in experimental conditions (see for example Barsanti et al., ACP 2013). The Griffin et al., 1999 parameters were chosen because of their use in the EPA chemical-transport model CMAQ. The Griffin et al., 1999 parameters represent a two-product model approach; to illustrate that different parameterizations will give different approximations, a set of VBS parameters was also used. We are not making any claims about the contribution of monoterpenes to BB-derived SOA relative to other BB-derived compounds. We are only demonstrating that for this given emissions source, and this given class of compounds, predictions of SOA may be significantly underestimated using alpha-pinene or "average" terpene SOA yields when camphene represents a significant fraction of the emissions (e.g., black spruce and Douglas fir).

**Comment 2.6:** One key aspect that this study fails to highlight – but one that is quite powerful - is the general approach to thinking about SOA formation from unexplored VOCs. I would recommend that a revision of this manuscript highlight this aspect. Particularly, since camphene, regardless of how it is dealt with, may not be important enough to meaningfully affect the total SOA burden from vegetation or biomass burning

**Response 2.6:** We agree! This is beyond the scope of this manuscript, but it is the focus of a subsequent manuscript that is in preparation.

**Minor Comments**

**Comment 2.7:** Line 53: Reaction rates or reaction rate constants? Having very different reaction rate constants doesn't necessarily mean that the atmospheric lifetimes could vary by the same amount.

**Response 2.7:** We agree. "Reaction rates" was changed to "reaction rate constants" and the text revised in lines 53-55:

*"The reaction rate constants of monoterpenes with atmospheric oxidants vary by orders of magnitude (Atkinson and Arey, 2003a; Geron et al., 2000), and their atmospheric lifetimes vary from minutes to days (Atkinson and Arey, 2003b)."*

**Comment 2.8:** Introduction: Well-cited section but could perhaps also be motivated by how these monoterpene emissions and their composition could change with future temperatures.

**Response 2.8:** This is a very good point, though currently, we don't think that there is enough known about how such emissions will change in the future. While there have been many efforts to quantify this change, it has often been in the absence of other expected changes (e.g., levels of $CO_2$, ozone, etc.), which also can affect emission rates.

**Comment 2.9:** Line 86: 'compared'.

**Response 2.9:** This has been corrected.

**Comment 2.10:** Section 2.2: How are the generalized reaction schemes for monoterpenes determined?

**Response 2.10:** A brief description of the generation of reaction schemes using GECKO-A has been in lines 128-134.

**Comment 2.11:** Line 186: Why was 50 degrees selected?

**Response 2.11:** The simulation conditions were selected to represent mean typical chamber. Mean photolysis conditions were thus calculated using a zenith angle of $50^o$. The details have been added to the sentence line 202-203:

*"...the solar zenith angle, required to calculate photolysis frequencies, was set at $50^o$ to represent mean daytime solar spectra."*

**Comment 2.12:** For the results discussed in Section 3.1, can time series plots for SOA mass, O:C, and major components be included in the main text or SI for the simulations?

**Response 2.12:** Time series plots for SOA mass, O:C, and top 10 products now have been included in the SI.

**Comment 2.13:** Table 3: This only shows modeled O:C values. Can a comparison similar to Figure 4 be also be performed for SOA O:C, in addition to or instead of the comparison to literature values in the second paragraph in Section 3.1.1? On a related note, HOM formation will tend to further increase the modeled O:C and further separate the modeled O:C from the literature data. Thoughts on why this might be? Is this related to not having enough semi-volatile material with low O:C to 'dilute' the bulk O:C?

**Response 2.12:** As noted above, we do not expect HOM formation/dimerization to contribute significantly to the final SOA mass/yield observed in these studies. Though we do agree that if such compounds were important in the dark ozonolysis studies, this would further separate the

measured and modeled O:C ratios. The literature-based observed O:C ratios, where available, have now been included in Table 3.

**Comment 2.14:** Line 225: It would seem like a lot of $H_2O$ would need to be lost (and lot of oligomers produced in the process) to explain the differences in the modeled and measured O:C. Can this calculation be done to test plausibility?

**Response 2.14:** This is a good point. The recommended calculation indicates that ~50% of mass would need to be oligomers to explain the difference in O:C. Prior GECKOA modeling-chamber comparisons (McVay et al., 2016) indicated that SOA yields increased significantly when accretion reactions were included, particularly when the faster rate constant was used. It is likely that the addition of accretion reactions would improve O:C agreements, but then may also result in overprediction of SOA mass relative to the chamber experiments (particularly for limonene, and without considering some loss processes). We have modified the sentence slightly to reflect the possibility but also uncertainty (lines 243-244):

*"Therefore, the lower observed O/C values may be partially explained by the loss of H2O during oligomerization, . . ."*

**Comment 2.15:** Line 227: It should have been relatively easy to zero out the OH in the model simulations to appropriately represent the chamber experiments where an OH scavenger was used.

**Response 2.15:** As suggested by the reviewer, sensitivity studies were run in which CO was used as an OH scrubber. The predicted O:C ratios were not significantly different between the sets of simulations, and thus this sentence has been removed.

**Comment 2.16:** Section 3.1.2: The volatility distribution of α-pinene SOA has been well studied (e.g., Saha and Grieshop, ES&T, 2016; Yli-Juuti et al., GRL, 2017) and to some degree its composition too (e.g., Sato et al., ACP, 2018; D'Ambro et al., ES&T, 2018). While a qualitative comparison has been done here, can this be done more quantitatively against literature data? For instance, could the modeled SOA products be lumped by volatility and compared against volatility distributions constructed from analyzing dilution, thermodenuder, and speciation data? Can the modeled composition be compared directly to measured data at the species level? This would help improve confidence in the model predictions.

**Response 2.16:** Quantitative comparisons of GECKO-A simulations with α-pinene chamber experiments for the purpose of SOA formation has been performed by Valorso et al., 2011; Denjean et al., 2015; McVay et al.,2016. These studies mainly focus on model/measurement comparisons of SOA mass and/or O:C ratio. A quantitative model/measure comparison on volatility distributions and/or gaseous and particulate individual species would indeed provide additional confidence in the model simulation, and allow to improve our knowledge of processes in particular those leading to HOM formation. Such a quantitative comparison can for sure be the objective of a paper but is out of the scope of the study presented here.

**Comment 2.17:** Line 254: Should it be 'macroscopic'?

**Response 2.17:** This has been corrected.

**Comment 2.18:** Line 263: Do you see an effect of photolysis on SOA mass yields? This recent study might be of interest to examine for consistency with this work: https://doi.org/10.1021/acs.est.9b07051

**Response 2.18:** We agree with the reviewer that representing loss processes are important, particularly when trying to match ambient observations or make atmospheric predictions. Gas-phase photolysis is represented in GECKO-A (including the current work), and particle-phase photolysis in GECKO-A has been explored by Hodzic et al. 2015. That said, the consideration of loss processes, including particle-phase photolysis, is beyond the scope of this work.

**Comment 2.19:** Line 288: Were these values picked to replicate atmospheric conditions? If yes, this should be stated. Was there a reason the model was not used to explore dependence on T, RH, NOx, etc?

**Response 2.19:** As noted in the manuscript, the controlled reactivity conditions represent conditions in which the gas-phase chemistry is not controlled by the individual hydrocarbons, as is true in the atmosphere. However, this is not the same as being relevant for atmospheric conditions, which cover large ranges in terms of the controlling variables. An exhaustive sensitivity study of the formation of camphene SOA as a function of all of the controlling variables is beyond the scope of this work.

**Comment 2.20:** Section 3.2: If SOA production was assumed to be instantaneous, would the kinetics under low condensational sink conditions, alter the model predictions significantly under atmospheric conditions?

**Response 2.20:** The assumption of instantaneous equilibrium would not likely affect SOA predictions under atmospheric conditions unless the temperature/relative humidity/OA composition were such that the particles were viscous. It is under such conditions, or for consideration of new particle formation, that SOA formation needs to be treated kinetically.

**Comment 2.21:** Section 3.2.3: How would HOM production and differences between the three VOCs affect the modeled particle-phase product distribution and its volatility? This should have a reasonable impact on the results shown in Figure 11.

**Response 2.21:** While the overall mass loadings are low enough in the controlled-reactivity simulations that HOM formation/dimerization could have a non-negligible contribution to the particle phase composition, the relatively short lifetimes of $RO_2$ (< 60 s) in these simulations suggest that HOM formation is relatively unimportant. In the event that HOM formation did occur, since HOM yields for limonene are typically higher than alpha-pinene, the difference in the % of E/LVOCs contributing to the particle-phase would be further magnified. No HOM yields have been published for camphene so it is unclear how the particle-phase volatility distribution would change, should HOM formation occur.

**Comment 2.22:** Line 375: 'lower than' what?

**Response 2.22:** This has been corrected and now reads (line 399):

*". . .lower than oxidation products predicted for α-pinene and limonene."*

**Comment 2.23:** Line 380: Needs to be made clear that this is only for the SOA contribution from monoterpenes, which could be quite small, depending on the fuel type.

**Response 2.23:** Agreed. The clause "from monoterpenes" has been added to line 404:

*"...increased predicted SOA mass from monoterpenes by 43-108 %."*

---

## Author Response (AR2)

**Responses to Reviewers (acp-2020-829)**

**Manuscript: "Using GECKO-A to derive mechanistic understanding of SOA formation from the ubiquitous but understudied camphene"**

We thank the reviewers and the editor for their comments on our revised submission. We believe we have addressed the outstanding concerns. Responses are in italics, modified text in red, and the line numbers refer to the "track changes" version of the revised manuscript.

Reviewer Report #1:

*No response needed.*

Reviewer Report #3:

I share the same concern as the Reviewer #2 pointed out that the modeling results are likely not representative of the actual atmosphere scenarios without carefully accounting for the autooxidation chemistry of peroxy radicals and their self/cross-reactions as a widespread source of highly oxidized monomeric and dimeric products (HOMs) that significantly contribute to the SOA formation in the monoterpene system. Recent kinetic studies have demonstrated that this autooxidation pathway could effectively operate and affect the product distributions from the ozonolysis of a-pinene at relatively short $RO_2$ bimolecular lifetimes, e.g., in the presence of ~ppb levels of NO (Lyer et al. Nature Communications 2021). This level of NOx as an upper bound below which the $RO_2$ isomerization outcompete their bimolecular reactions with NO/RO2/HO2 is relevant to the conditions vastly encountered in most regions of the U.S. and also those conditions measured in fire plumes.

*One of the major objectives of this work was to better understand the chemistry of camphene; particularly in comparison to more well-studied monoterpenes, alpha-pinene and limonene. The controlled reactivity simulations were designed to facilitate comparison of these three terpenes in a buffered system, in which the gas-phase oxidant levels were controlled and the $RO_2$ radicals reacted equally with $HO_2$ and NO. We recognize that this does not cover the suite of chemical conditions in the ambient atmosphere, including those in which unimolecular $RO_2$ reactions play a critical role in forming SOA and defining SOA composition and properties. We believe that the interesting results of this first chemically-detailed modeling study will encourage more complete characterization of camphene gas-phase chemistry and SOA formation under a range of atmospherically relevant conditions. To further emphasize these points, the following edits have been made:*

*line 22: added "and peroxy radical reacted equally with HO2 and NO".*

*line 33-34: revised to read "This first detailed modeling study of the gas-phase oxidation of camphene and subsequent SOA formation highlights opportunities for future measurement-model comparisons and lays a foundation for developing chemical mechanisms and SOA parameterizations for camphene that are suitable for air quality modeling."*

*lines 437-440: To the closing sentence, "Further modeling and/or experimental studies are needed to develop and test a suitable SOA parameterization for representing camphene in air quality models", we added the clause: "including a robust assessment of the role of gas-phase HOM formation via $RO_2$ autooxidation, and condensed-phase accretion reactions, on SOA composition and yields under a range of atmospherically relevant conditions."*

*We agree with the reviewer that the understanding of the conditions under which unimolecular $RO_2$ reactions can occur in the ambient atmosphere is expanding, and that accounting for HOM formation by unimolecular $RO_2$ reactions, and subsequent dimerization, will be critical for accurate predictions of SOA formation and composition in the ambient atmosphere. We note that while such reactions are not considered in this work, given the recent availability of SARs to predict the rate coefficients of $RO_2$ H-shifts, the product distributions presented in the mechanism schematics may be useful for understanding the relative or potential importance of such reactions in the systems studied.*

While I understand, as the authors stated, that full consideration of HOM formation/dimerization is not possible at this time, I suggest, however, a number of sensitivity tests need to be at least performed to evaluate the uncertainties arising from this missing chemical mechanism in the simulations. My understanding that such tests are feasible based on previous published GECKO-A studies, see a couple of examples below.

*One of the strengths of a near-explicit chemical mechanism model is that parameterized representations of key reactions are not necessary, provided relevant data and/or SARS exist. Given the recent availability of the referenced SAR by Vereecken and Noziere, we believe that a more rigorous assessment and greater contribution can be made by incorporating a version of that SAR into GECKO-A and running camphene simulations under a range of atmospherically relevant scenarios. We have added text throughout the manuscript to more clearly acknowledge the lack of HOM monomer formation via $RO_2$ autoxidation, and the implications for the results and conclusions presented.*

*line 103-109: revised to read:* "*Autoxidation, leading to the formation of highly oxygenated molecules (HOM) in the gas phase (e.g., Bianchi et al., 2019; Ehn et al., 2014), is not currently represented in GECKO-A. A SAR to predict the rate coefficients of peroxy radical ($RO_2$) H migration reactions (H-shifts) that lead to the formation of HOM was recently published by Vereecken and Nozière (2020). The straight implementation of this SAR into GECKO-A would lead to a non-manageable number of species and reactions.* Therefore, *reduction protocols are currently under development to consider the autoxidation reactions in subsequent model versions. For the application presented herein, limitations and implications of the absence of HOM formation via $RO_2$ autoxidation are discussed where relevant*."

*lines 190-192: Added,* "*It is noted that the simulations are unable to capture HOM formation via $RO_2$ autoxidation and subsequent dimerization (Ehn et al., 2014), that may have occurred in the chamber studies, particularly under DO conditions.*"

*lines 232-243: Added,* "*Overall, the model simulations agree well with the observed trends in SOA yield as a function of SOA mass. The largest discrepancies are for α-pinene ozonolysis, in which SOA mass is underpredicted relative to the observations. The contribution of HOM formation from $RO_2$ autoxidation is expected to be more important under such conditions, when the lifetime of $RO_2$ is sufficiently long for autoxidation to compete with biomolecular reactions and monoterpene oxidation by $O_3$ is greater than by OH leading to higher HOM yields (Ehn et al., 2014; Jokinen et al., 2015). The inclusion of HOM formation and subsequent dimerization would lead to an increase in predicted SOA mass in both the α-pinene and limonene ozonolysis simulations. An increase in SOA mass due to HOM formation and subsequent dimerization would improve the measurement-model agreement for α-pinene, but would also lead to an overprediction of SOA mass for limonene. In addition, a non-negligible contribution of HOM monomers and dimers to the particle phase would increase the calculated O/C ratio, and increase the measurement-model discrepancy further discussed below. McVay et al. (2016) reported similar conclusions for α-pinene photolysis experiments; a parameterized representation of $RO_2$ autoxidation in GECKO-A increased predicted SOA mass for low UV conditions, improving measurement-model agreement at the end of the experiment; and resulted in no change for high UV conditions.*"

*lines 332-334: Added, "The calculated lifetime of $RO_2$ with $HO_2$/NO was < 60 s, and thus it is assumed that these biomolecular $RO_2$ reactions would be dominant, and the absence of HOM formation via $RO_2$ autooxidation in GECKO-A did not significantly impact the results and conclusions derived from these simulations."*

1. The authors could refer to McVay et al. ACP (2015) in terms of adding the RO2 isomerization channel to the original GECKO mechanism. Recent published RO2 isomerization kinetics in a-pinene ozonolysis (e.g., Kurteń et al., JPCA, 2015; Zhao et al., PNAS, 2018) could be adapted to assess the relevance of the autooxidation chemistry under conditions simulated in this study and how this chemistry could change the product distribution and consequently the SOA yields and composition.

*In McVay et al., HOM formation via autooxidation was represented by adding a single product from the alpha-pinene + $O_3$ reaction with a molar yield of 7%. The results of this representation were variable in the context of improving measurement-model agreement. This study has now been explicitly referenced in the context of including a parameterized representation of HOM formation in GECKO-A (see red text above).*

2. La et al. ACP (2016) incorporated a heterogeneous reaction pathway in the GECKO simulations of the SOA formation from photooxidation along-chain alkanes. For this study, it is important to test how the particle-phase dimerization such as the peroxyhemiacetal formation from the poly-peroxides that are largely present in the HOMs molecules (see kinetics in e.g., Bakker-Arkema and Zimemann 2020) could alter the SOA mass and composition.

*In the manuscript, we differentiate HOM formation via gas phase $RO_2$ autooxidation (as defined by Bianchi et al. 2019) from accretion product formation via heterogeneous or condensed-phase reactions. Highly-oxygenated (6+ O) gas-phase products are formed in the GECKO-A modeling simulations, but as noted in the manuscript and reviewer responses, $RO_2$ autooxidation and condensed-phase reactions are not represented. We agree with the reviewer that accretion product formation is important, and we acknowledge in the manuscript that such reactions are likely occurring in the chamber studies. We have added text throughout the manuscript to more clearly acknowledge the lack of condensed-phase reaction chemistry, and the implications for the results and conclusions presented.*

*line 125: Revised to read, "Condensed-phase reactions are not currently represented in GECKO-A; the limitations and implications of which are discussed where relevant."*

*lines 377-380: Revised to read, "Product volatility distributions can be influenced by gas-phase $RO_2$ autooxidation, and condensed-phase reactions, which were not considered here. While HOM formation likely played a minor role in these controlled reactivity simulations, the monomer building blocks of known accretion reactions were predicted for all monoterpenes studied. Thus, it is expected that accretion product formation could occur under these conditions, leading to changes in the simulated volatility distributions."*

*lines 430-431: Added, "The predicted SOA yields do not account for condensed-phase accretion reactions, which could occur under the simulation conditions."*